# Chain of Agents: Large Language Models Collaborating on Long-Context Tasks

**Yusen Zhang**♣*, **Ruoxi Sun**◇, **Yanfei Chen**◇, **Tomas Pfister**◇, **Rui Zhang**♣†, **Sercan Ö. Arık**◇†
♣ Penn State University, ◇ Google Cloud AI Research
{yfz5488, rmz5227}@psu.edu, {ruoxis, yanfeichen, tpfister, soarik}@google.com

## Abstract

Addressing the challenge of effectively processing long contexts has become a critical issue for Large Language Models (LLMs). Two common strategies have emerged: 1) reducing the input length, such as retrieving relevant chunks by Retrieval-Augmented Generation (RAG), and 2) expanding the context window limit of LLMs. However, both strategies have drawbacks: input reduction has no guarantee of covering the part with needed information, while window extension struggles with focusing on the pertinent information for solving the task. To mitigate these limitations, we propose *Chain-of-Agents (CoA)*, a novel framework that harnesses multi-agent collaboration through natural language to enable information aggregation and context reasoning across various LLMs over long-context tasks. CoA consists of multiple worker agents who sequentially communicate to handle different segmented portions of the text, followed by a manager agent who synthesizes these contributions into a coherent final output. CoA processes the entire input by interleaving reading and reasoning, and it mitigates long context focus issues by assigning each agent a short context. We perform a comprehensive evaluation of CoA on a wide range of long-context tasks in question answering, summarization, and code completion, demonstrating significant improvements by up to 10% over strong baselines of RAG, Full-Context, and multi-agent LLMs.

## 1 Introduction

Despite their impressive performance across a wide range of scenarios, LLMs struggle with tasks that involve long contexts [8, 63, 57]. Numerous application scenarios demand extremely long contexts, such as question answering [85, 22, 69], document and dialogue summarization [25, 93, 91, 90, 12], and code completion [20, 43], where the inputs contain entire books [32, 33] and long articles [16].

To tackle the challenge with long context tasks, two major directions have been explored as shown in Table 1: *input reduction* and *window extension*. *Input reduction* reduces the length of the input context before feeding to downstream LLMs. Truncation approaches [1, 67] directly truncate the input. Retrieval Augmented Generation (RAG) [81] extends this direction by retrieving the most relevant chunks through embedding similarity. However, because of low retrieval accuracy, LLMs could receive an incomplete context for solving the task, hurting performance. *Window extension* extends the context window of LLMs via finetuning to consume the whole input [13, 44, 48]. For example, Claude-3 [5] directly allows reading 200k tokens for each input. However, when the window becomes longer, LLMs struggle to focus on the needed information to solve the task, suffering from ineffective context utilization such as the *"lost in the middle"* issue [37, 3, 42].

---

*Work done while the author was a student researcher at Google Cloud AI Research.
†Last authors

38th Conference on Neural Information Processing Systems (NeurIPS 2024).

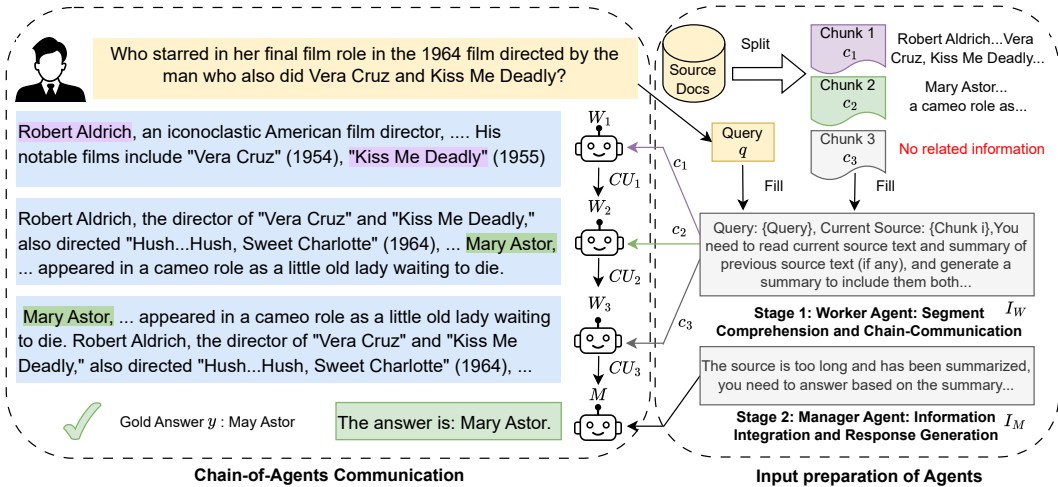

Figure 1: Overview of Chain-of-Agents, a training-free, task agnostic, and highly-interpretable framework that harnesses multi-agent collaboration for long-context tasks. Blue boxes indicate communication unit $CU_i$ between worker agents $W_i$ and $W_{i+1}$. It consists of multiple worker agents who sequentially communicate to handle different segmented portions of the text, followed by a manager agent who synthesizes these contributions into a coherent final output.

Table 1: Comparison between Chain-of-Agents and prior methods for long-context tasks. Rec./Foc.: being able to mitigate inaccurate receptive field/long context focusing issues. Read: the number of tokens as model input, where $n$ is the total input length, $k$ is the context window limit of LLMs. Inter.: the interpretability of the approach. Note that RAG is 'medium interpretable' because of the re-ranked chunks.

| Category | Example Work | Rec. | Foc. | No Train | Read | Agent | Applicability | Inter. |
|----------|--------------|------|------|----------|------|-------|---------------|--------|
| Input Reduction | Truncation [50] | ✗ | ✓ | ✓ | $k$ | Single | Generic | Low |
| | RAG [81] | ✗ | ✓ | ✗ | $n + k$ | Single | Query-based | Medium |
| Window Extension | Position Interpolation [13] | ✓ | ✗ | ✗ | $n$ | Single | Generic | Low |
| | Long Context [5] | ✓ | ✗ | ✗ | $n$ | Single | Generic | Low |
| Multi-agent LLMs | Chain-of-Agents (Ours) | ✓ | ✓ | ✓ | $n$ | Multiple | Generic | High |

Motivated by the aforementioned challenges, we propose a novel framework, *Chain-of-Agents (CoA)*, inspired by the way humans interleave reading and processing of long contexts under limited working memory constraints of brains [15]. The key idea of CoA is to harness multi-agent communication to enable information aggregation and context reasoning capabilities across different LLMs. As shown in Figure 1, CoA contains two stages. In stage 1, a series of worker agents in charge of different chunks of long context collaborate and aggregate evidence for answering the given query. To this end, the workers read and process sequentially, each receiving the message from previous worker and transferring the useful updated information to the next. In stage 2, the manager agent receives the complete evidence from last worker agent and generates the final response.

As shown in Table 1, CoA is a training free, task agnostic, and highly interpretable framework processing entire "receptive field" by interleaved reading-processing and mitigating the long context focusing issue by assigning each agent a short context. Different from *input reduction* where LLMs need to start processing with low receptive field over reduced inputs ("read-then-process"), workers in CoA start to process each chunk before reading all input ("interleaved read-process"), tackling the problems that input reduction struggles with, such as, generic summarization or counting of passages [6]. Different from *context extension*, CoA leverages the capability of communication rather than trying to feed many tokens into an LLM. This is a more natural solution for complex tasks because we assume that each LLM has its limit and there are always complex context tasks surpassing its limit. Compared with Full-Context, CoA is also cost effective by reducing time complexity from $n^2$ to $nk$, where $n$ is input tokens and $k$ is the context limit of LLMs.

We conduct intensive experiments on *nine datasets*, including question answering, summarization, and code completion tasks with *six LLMs*, with PaLM 2 [4], Gemini [67], and Claude 3 [5] models. We compare CoA with two strong baselines chosen from *input reduction* and *window extension* approaches, respectively: (i) RAG, which uses a state-of-the-art retriever to obtain the most relevant information to feed into the LLM and (ii) Full-Context (Vanilla), which feeds all input into the LLM until reaching the window limit. Our results show that on all nine datasets, CoA obtains significant improvement over all baselines by up to 10%. Noting that there is not enough research on multi-agent for long context tasks, we carefully create two multi-agent baselines, including a hierarchical structure and result merging approach to further demonstrate that CoA is superior among other possible multi-agent frameworks.

## 2 Related work

**Multi-agent LLMs.** Multi-agent LLMs has become a popular topic [21]. A large proportion of works focus on social simulation [52, 53] and frameworks [35]. Based on the success of them, some works explore the game settings [39, 74, 82, 83, 47], world wars [23], economy markets [36, 76], recommendation systems [88], and pandemics [19]. Others advance problem solving, focusing on reasoning of short text via multi-agent debating [18, 80, 10, 66] and discussing [11, 58] for different tasks in reasoning [18, 66], mechanics problems [49], paper review [84], knowledge graph construction [86], and code intelligence [70, 24]. Different from the above works, we improve problem-solving on long context tasks. To the best of our knowledge, the closest work LongAgent [92] utilizes a tree structure to do multi-hop QA over a long input context and the sibling worker agents do not communicate, while CoA utilizes a chain structure allowing the communication units to flow between workers (detailed comparison in Appendix G and F.3).

**Long Context Modeling for LLMs.** *Input Reduction*: RAG is broadly leveraged to solve long context query-based tasks [81, 2]. Combined with a strong retriever [79, 41, 73], LLMs are expected to handle long context questions in open domains. Previous studies have augmented LLMs during pretraining [26, 72], finetuning [34], inference [87], or directly integrating [28, 61]. Moreover, some token-level retrieval approaches are proposed [38]. Longllmlingua [27] removes tokens from long prompt to compress long context prompt to a desired budget. *Window Extension*: The context windows of LLMs are getting longer and longer thanks to the development of GPUs. For instance, the context window increases from 1024 (GPT-2 [56]), 2048 (GPT-3 [7]), to 128k (GPT-4 [50]). Moreover, the newest version of Claude-3 [5] supports 200k context windows. To save the cost of LLM training, some continual learning or finetuning approaches are proposed to extend the context window of pretrained LLMs [46, 55, 44, 48]. For instance, position interpolation [13] modifies rotary position encoding [64] and extends the context length of LLaMA [68] to 32k. Different from the above works, CoA does not reduce the input length or extend the window length of LLMs, but rather leverages multi-agent collaboration and communication to obtain the full receptive field.

Recently, a few studies have employed text chunking algorithms to divide and process lengthy text inputs. Notably, RecurrentGPT [95] adopts a chain structure to generate long outputs in a chunk-by-chunk manner, maintaining continuity in the storyline. WalkMaze [9] utilizes a centralized tree structure without communication between sibling agents to do single-hop QA over a long input context. Different from these works, CoA focuses on long input tasks with a chain structure with inter-sibling communication (Append G).

**Complex Task Reasoning.** Previous works on complex reasoning have focused on decomposing the complex question into sub-questions to solve them step-by-step. [54] decompose the questions with an unsupervised model and answer them separately with another model. Decomposed Prompting [30] leverages some predefined modules to classify each decomposed sub-question, then further decompose if needed. Additionally, decomposing is used for human-computer interaction [77], and prompter training [71]. Recently, many work has been proposed for LLMs, such as Chain-of-thought [75] Least-to-most prompting [94] and Pearl [65]. However, the length of the prompt does not exceed the context limit of a single agent. By contrast, our Chain of Agents framework is proposed to effectively reason across multiple agents to support the unlimited length of source text.

## 3 Method

Figure 1 shows the overview of our Chain-of-Agents (CoA) framework, containing two stages. In stage 1, long context is split into chunks where each chunk can be processed by a worker agent. Then, the worker agents communicate sequentially to produce evidence over the entire context. In stage 2, a manager agent consumes the knowledge from the chain of workers to generate the final answer.

To formulate the task, we denote a long-context sample as $(x, y, q)$, where $x$ is the input of $n$ tokens, $y$ is the output of $m$ tokens, $q$ is an optional query. Given an LLM with $k$ tokens (usually $k \ll n$) as the context window limit, the target is to generate $y$ with the limited input context window. Therefore, we divide each source text $x$ into chunks $x = \{c_1, c_2...c_l\}$, so that each chunk can be completely fed into the LLM agent backbone model.

---

**Algorithm 1** Chain of Agents (CoA).

---

**Input:** Source input $x$, query $q$, agent window size $k$,
  large language model LLM($*$).
**Output:** Answer to the query.
  Split $x$ into $l$ chunks $\{c_1, c_2, \cdots, c_l\}$
  where $c_i$ is shorter than $k$
  Initialize $CU_0 \leftarrow$ empty string.
  **for** $i$ in $1, 2, \cdots, l$ **do**
    $CU_i \leftarrow \text{LLM}_{W_i}(I_W, CU_{i-1}, c_i, q)$
  **end for**
  **return** $\text{LLM}_M(I_M, CU_l, q)$

---

### 3.1 Stage 1: Worker Agent: Segment Comprehension and Chain-Communication

In Stage 1, CoA contains a sequence of $l$ number of worker agents. Each worker $W_i$ inputs the concatenation of a chunk $c_i$ from source text $x$, a query $q$, instruction for a specific task for worker agent $I_W$, and the message passed from the previous agent, denoted as "communication unit" $CU_{i-1}$. The communication is unidirectional, passing from worker agent $i - 1$ to $i$. The worker agents process them and output the message $CU_i$ for next worker, expressed as:

$$CU_i = \text{LLM}_{W_i}(I_W, CU_{i-1}, c_i, q), \tag{1}$$

CUs produced by worker agents vary across different tasks. For question answering, CU contains the evidence for the manager to answer the question. For summarization, CU contains the summary of the previous texts. For code completion, CU contains the code summary with function/class names and explanation. Effectiveness on diverse tasks demonstrates the flexibility of CoA (Appendix C).

The multi-step worker communication in CoA expands the model context to the full receptive field, meaning that the last worker can read the full input no matter how long the input is. Therefore, CoA is extensible to inputs with different lengths by adjusting the number of worker agents.

The left side of Figure 1 underscores the necessity of collaborative communication among workers to effectively address complex, long-context reasoning tasks. We observe that 1) Although the question is unanswerable given $c_1$, $W_1$ generates related evidence that is useful for answering the question; 2) with the partial answer from the previous worker, $W_2$ further reasons with the current source to complete the full reasoning chain across agents and generate the interpretative reasoning chain; 3) $W_3$ finds no related information in the chunk 3, it directly rewrites $CU_2$ by putting the correct answer as the first token of $CU_3$ without adding any unrelated information (Appendix F.4). This shows that if workers are independent (such as tree structure communication), it is impossible to answer hop two while the answer of hop one is held by another worker (Appendix G).

### 3.2 Stage 2: Manager Agent: Information Integration and Response Generation

In Stage 2, after multiple steps of information extraction and comprehension by worker agents, the manager agent produces the final solution. While worker agents extract relevant information in a long-context source, the manager agent synthesizes relevant information accumulated by the end of *"worker-agent-chain"* to generate the final answer. Specifically, given the instruction for manager $I_M$ and query $q$, the manager agent consumes accumulated knowledge from last worker $CU_l$ and generates the final answer $Response$:

$$Response = \text{LLM}_M(I_M, CU_l, q) \tag{2}$$

The benefit of using a separate LLM as the manager agent is to decompose the duty of analyzing chunks in the long-context source (*"worker agents"*) and producing the final answer (*"manager agent"*), so that every agent can fulfill its duty to the most[3].

---

[3]Other design choices: Our experiments show that using the last worker $W_l$ to directly generate the final result leads to a performance drop. Besides, feeding the manager with all $CU_i$ or some $CU$ that is related to the answer (decided by $W_i$) also hurts the performance because of confusion led by conflicting $CU_i$.

Table 3: **Dataset Statistics**. Avg. Input/Agents is the average words/agents (8k) for source input.

| | Question Answering | | | | | Summarization | | | Code |
| --- | --- | --- | --- | --- | --- | --- | --- | --- | --- |
| | HotpotQA | MuSiQue | NarrativeQA | Qasper | QuALITY | QMSum | GovReport | BookSum | RepoBench-P |
| Avg. Input | 10603 | 12975 | 71787 | 4236 | 4936 | 12524 | 9239 | 108478 | 7105 |
| Avg. Agents | 2.35 | 2.88 | 12.45 | 1.12 | 1.31 | 2.57 | 2.03 | 18.63 | 1.69 |
| Query-based | ✓ | ✓ | ✓ | ✓ | ✓ | ✓ | ✗ | ✗ | ✓ |

## 3.3 Time Complexity Analysis

We compare the time cost of full-context input and Chain-of-Agents theoretically in a decoder-only setting. We assume the response generated by LLMs contains $r$ tokens on average, the input has $n$ tokens, the context limit of LLM is $k$, and the length of each chunk in RAG is $k'$. The time complexity is shown in Table 2 (Ap-

Table 2: Time complexity.

| | Encode | Decode |
| --- | --- | --- |
| Full-Context | $\mathcal{O}(n^2)$ | $\mathcal{O}(nr)$ |
| CoA | $\mathcal{O}(nk)$ | $\mathcal{O}(nr)$ |
| RAG | $\mathcal{O}(nk') + \mathcal{O}(k^2)$ | $\mathcal{O}(n/k') + \mathcal{O}(kr)$ |

pendix A). As can be seen, the encoding time of CoA is less than Full-Context because $k \ll n$ in long context tasks, while they have the same decoding time. This demonstrates the efficiency of CoA compared with the Full-Context baseline.

## 4 Experiment

### 4.1 Experiment Setup

**Datasets.** We conduct experiments on nine long context datasets across three task types (Table 3):

- **Question Answering.** We consider five QA datasets from the LongBench [6] and SCROLL [60]. **HotpotQA** [85] is a Wikipedia-based multi-hop QA dataset. It requires reasoning across multiple passages to find the answer. **MuSiQue** [69] is a multi-hop QA dataset. It is much more difficult than HotpotQA as it contains more hops in one sample, unanswerable questions, and harder distracting content. **NarrativeQA** [31] is a QA dataset over entire books or movie transcripts. The answers can be abstract or extractive, yes/no, and unanswerable. **Qasper** [17] is a question answering dataset over NLP papers. It also contains extractive, abstractive, yes/no, and unanswerable questions. **QuALITY** [51] is a dataset based on stories and articles with multiple-choice questions for each sample. The model needs to select the correct answer among choices.
- **Summarization.** We pick two summarization datasets from SCROLLS. **QMSum** [93] is a query-based summarization dataset, formed by meeting transcripts from multiple domains such as academic and industrial products. **GovReport** [25] is a generic summarization dataset containing long reports published by the U.S. Government Accountability Office. We also use one dataset for long context memorization tasks. **BookSum** [33] is a collection of datasets for long-form narrative summarization, including novels, plays, and stories. We use the book-level partition of the BookSum dataset for experiments.
- **Code Completion.** We pick **RepoBench-P** [43] which is collected from GitHub repositories, and the model needs to generate the next line of code given the long code base.

**Metrics.** We report the geometric mean of ROUGE [40] for Summarization tasks, code similarity score [6] for Code Completion task, exact match for QuALITY dataset [60], and F1 score for the rest of the Question Answering datasets [6].

**LLMs.** We use six LLMs in total as the backbone of CoA across all experiments (Appendix B). **PaLM 2** [4] is a series of models with a dense left-to-right, decoder-only language model pretrained on a high-quality corpus of 780 billion tokens. We use **text-bison@001** and **text-unicorn@001** for the experiments with an 8k maximum context window. **Gemini 1.0** [67] is a family of LLMs proposed by Google. We use **gemini-ultra** for experiments. The input limit is 32k tokens for Gemini. **Claude 3** [5] is a family of large language models developed by Anthropic. The family includes three state-of-the-art models in ascending order of capability: **claude-3-haiku**, **claude-3-sonnet**, and **claude-3-opus**. These models are capable of consuming 200k tokens in the context window, providing a strong baseline for long context tasks. Although our framework is flexible to use diverse types of LLMs as workers and manager, we use the same model for each $W_i$ and $M$ if not specified.

Table 4: **Overall results of CoA**. CoA significantly outperforms Vanilla and RAG using various backbone LLMs on all datasets.

| LLMs | Baselines | Question Answering | | | | | Summarization | | Code |
|---|---|---|---|---|---|---|---|---|---|
| | | HotpotQA | MuSiQue | NarrativeQA | Qasper | QuALITY | QMSum | GovReport | RepoBench-P |
| text-bison | Vanilla (8k) | 45.57 | 26.87 | 11.96 | 26.56 | 61.86 | 15.45 | 20.60 | 56.30 |
| | RAG (8k) | 51.91 | 33.83 | 14.20 | 27.20 | 55.28 | 15.59 | 20.83 | 55.63 |
| | CoA (8k) | **53.62** | **37.09** | **25.26** | **37.17** | **65.42** | **16.77** | **26.11** | **58.25** |
| text-unicorn | Vanilla (8k) | 51.09 | 29.67 | 14.45 | 27.81 | 83.40 | 16.61 | 23.50 | 53.87 |
| | RAG (8k) | 58.01 | 40.38 | 19.12 | 24.44 | 83.00 | 16.83 | 21.43 | 50.49 |
| | CoA (8k) | **62.04** | **42.49** | **20.37** | **38.01** | **83.80** | **17.67** | **26.48** | **60.39** |
| gemini-ultra | Vanilla (8k) | 40.62 | 23.61 | 7.71 | 20.59 | 57.40 | 12.10 | 26.18 | 49.09 |
| | Vanilla (32k) | 45.09 | 27.93 | 7.21 | 21.71 | 58.60 | 10.24 | 26.96 | 73.04 |
| | RAG (8k) | 51.13 | 31.56 | 14.51 | 18.70 | 62.40 | 12.70 | 25.87 | 72.94 |
| | CoA (8k) | **54.26** | **35.09** | **25.26** | **35.10** | **80.60** | **12.84** | **26.98** | **73.05** |

**Baselines.** Our principle of choosing baselines is to find the strongest and most typical approaches from *input reduction* and *window extension*. The first baseline is **Vanilla**. It directly consumes tokens until the context window of LLM is fully utilized, implying a 200k window LLM if using Claude 3. The other one is Retrieval-Augmented Generation (**RAG**). We use the state-of-the-art retriever [79]. Following [81], we first segment the source text into chunks of 300 words and re-rank them using a retriever. Top-n chunks are then fed into the downstream LLM until the context window is fully utilized. GovReport dataset does not contain a query initially, we create a pseudo query "What is the summary of the whole government report?" as the query to rerank.

To evaluate the performance of CoA compared with possible multi-agent approaches, we carefully construct two approaches. Similar to CoA, we also assign each chunk $c_i$ to $W_i$ using similar instructions to generate $CU_i$. In these approaches, worker agents are parallel and independent while CoA is sequential. **Multi-Agent Voting** (**Merge**) Each agent directly generate an answer $a_i$ according to $c_i$. A majority voting is applied to all answers $a_i$ to decide the final answer. **Multi-Agent Hierarchical Structure** (**Hierarchical**). Inspired by [9], we propose a hierarchical framework, where the communication forms a tree structure between workers $W_i$ and manager $M$. For each worker, it first judges whether $c_i$ contains useful information. If true, it generates a communication unit $CU_i$. Then, all $CU_i$ are sent to the manager $M$ to come up with a final answer. Besides, we append an integer number $L$ at the end of every approach to clearly remind the window size limit of LLM. For instance, "CoA (8K)" refers to the base LLM used in CoA with window size 8K.

## 4.2 Overall Results of CoA

**Question Answering.** Table 4 shows the results of Question Answering tasks on all three models. CoA (8k) outperforms Vanilla (8k) by a large margin on *all 8 datasets*, including 13.30% on NarrativeQA, 12.82% on MuSiQue, and 22.00% on Quality, for text-bison, text-unicorn, and gemini-ultra, respectively. Also, CoA (8k) outperforms RAG (8k) model for all 8 datasets using all three LLMs, demonstrating that CoA achieves higher performance than RAG. In other words, **using multi-agent LLMs outperforms RAG models**. It is also worth noting that for gemini-ultra, Vanilla (32k) improves the Vanilla (8k) baseline, yet it is still lower than CoA (8k). CoA also obtains compatible performance with with state-of-the-art models (Appendix E), while keeping robustness of window size (Appendix F.2).

**Summarization and Code Completion.** Table 4 shows the results of Summarization and Code Completion tasks. Similarly, CoA (8k) also outperforms all Vanilla (8k) and (32k) baselines on all three datasets, demonstrating the strong capability of CoA on various tasks. It is worth noting that for GovReport, RAG fails to improve the baseline with pseudo query. By contrast, CoA improves the performance significantly, showing that **CoA can also be applied in non-query tasks** (Appendix F.1).

**Long Context LLMs.** As Claude 3 models support 200k of tokens, we call these models long context models (LCM). Table 5 shows the performance of the LCM on two datasets. As can be seen, CoA (8k) outperforms Vanilla (200k) significantly, showing that with only an 8k context window, **CoA achieves a much higher performance than LCM with a 200k context window**. Also, CoA improves the performance with the samples that can be fed into a 200k context window (no truncation). Moreover, the improvements over the Vanilla (200k) and RAG (8k) become higher when the model

Table 5: Comparison with long context LLMs on NarrativeQA and BookSum. CoA significantly outperforms Claude 3 with 200k context limits. No Trun./Trun. indicates the source text in the sample is less/more than 200k tokens which does not need/needs truncation for vanilla (200k) baseline. Average is the mean value across all samples.

| | claude-3-haiku | | | claude-3-sonnet | | | claude-3-opus | | |
|---|---|---|---|---|---|---|---|---|---|
| | No Trun. | Trun. | Average | No Trun. | Trun. | Average | No Trun. | Trun. | Average |
| **NarrativeQA** | | | | | | | | | |
| Vanilla (200k) | 8.00 | 2.50 | 7.17 | 5.58 | 2.44 | 5.15 | 7.23 | 2.35 | 6.56 |
| RAG (8k) | 5.94 | 4.22 | 5.71 | 9.09 | 5.17 | 8.50 | 6.13 | 4.29 | 5.86 |
| CoA (8k) | **18.31** | **21.34** | **18.80** | **16.63** | **16.47** | **16.51** | **24.38** | **21.26** | **23.96** |
| **BookSum** | | | | | | | | | |
| Vanilla (200k) | 11.98 | 11.70 | 12.04 | 12.17 | 11.90 | 12.10 | 14.11 | 13.67 | 14.00 |
| CoA (8k) | **13.28** | **13.73** | **13.70** | **14.92** | **15.05** | **14.96** | **17.74** | **16.68** | **17.47** |

Table 6: Comparison between CoA and other multi-agent frameworks. CoA with sequential agents outperforms other designs with multiple parallel agents including Merge and Hierarchical.

| | HotpotQA | MuSiQue | NarrativeQA | Qasper | QuALITY | QMSum | GovReport | RepoBench-P |
|---|---|---|---|---|---|---|---|---|
| Vanilla (8k) | 45.57 | 26.87 | 11.96 | 26.56 | 61.86 | 15.45 | 20.60 | 56.30 |
| Merge (8k) | 42.96 | 26.66 | 11.27 | 26.78 | 59.30 | 9.42 | 25.38 | 33.66 |
| Hierarchical (8k) | 50.62 | 29.40 | 17.04 | 31.39 | 64.20 | 15.19 | 16.54 | 27.96 |
| CoA (8k) | **53.62** | **37.09** | **25.26** | **37.17** | **65.42** | **16.77** | **26.11** | **58.25** |

size increases from Haiku to Opus (11.63/11.36/17.4 for NarrativeQA, 1.66/2.86/3.47 for BookSum). This demonstrates that **CoA benefits from stronger models to achieve higher improvements**.

**Other Multi-Agent Frameworks.** As shown in Table 6, Hierarchical (8k) outperforms Vanilla (8k) on five out of eight datasets, demonstrating the hierarchical approach can also improve the vanilla baselines significantly. Merge (8k) is lower than Vanilla (8k) except for GovReport, showing that merging is especially effective for long summarization tasks such as GovReport. As can be seen, CoA outperforms Hierarchical and Merge on all eight datasets. The reason behind the results is because Hierarchical and Merge do not allow workers to communicate with each other due to their parallel designs. Thus, each worker can only maintain the information in its own chunk $c_i$ which blocks the understanding of the whole text, hurting the performance greatly.

## 5 Analyses

### 5.1 CoA Improvement is More Obvious When RAG Fails to Retrieve Gold Answer

To demonstrate this, we first classify the samples in NarrativeQA dataset into different bins according to the position (index) of the chunk in RAG processed input that contains the gold answer. Then, we compute the average score of the CoA and RAG results of different bins. Figure 3 shows the results. As shown in the figure, RAG performs better when the index is smaller (the gold answer is nearer to the top), showing that downstream LLMs rely significantly on the quality of RAG re-ranking. Besides, the performance of RAG is positively correlated to CoA's when it successfully retrieves the gold answer. However, when RAG fails, CoA can greatly improve the performance (much higher than the tendency line).

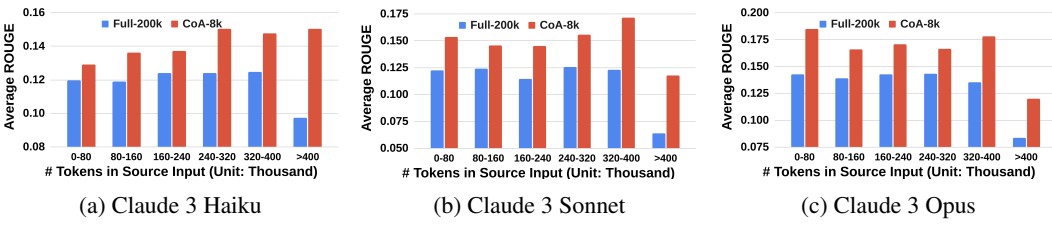

(a) Claude 3 Haiku          (b) Claude 3 Sonnet          (c) Claude 3 Opus

Figure 2: Performance of Claude 3 on BookSum. Improvement is more obvious for longer inputs.

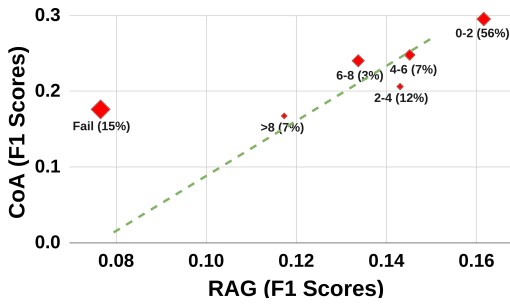

Figure 3: Comparison on NarrativeQA. X-axis/Y-axis indicate RAG/CoA performance while each point represents a bin. The number indicates the chunk index of gold answer (ratio of number of samples in bracket), and the size of the point indicates the improvement of CoA over RAG.

Figure 4: Performance of CoA and Full on Natural Questions. CoA mitigates the lost-in-the-middle issue. X-axis is the index of document with gold answer where small number indicates gold answer is closer to start.

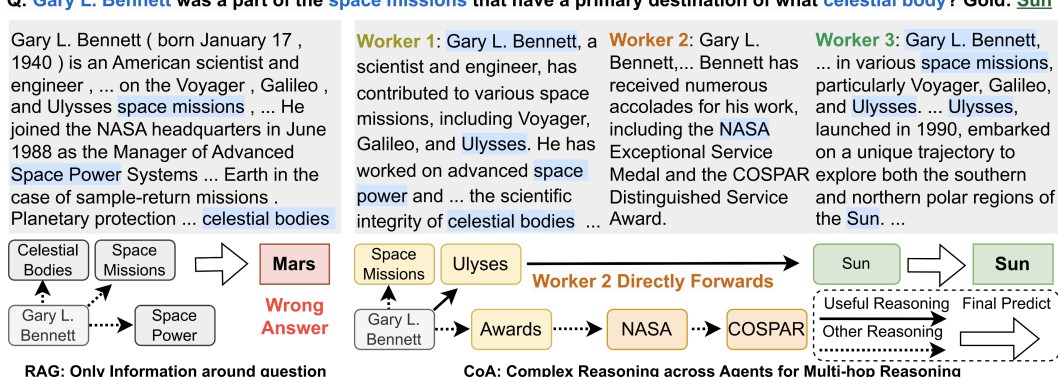

Figure 5: A case study of RAG (left) and CoA (right) on HotpotQA. The sequential agent communication enables CoA to perform complex multi-hop reasoning over long contexts.

## 5.2 CoA Improvement is More Obvious When Long Context Models Meet Longer Inputs

We compare the performance of CoA and Vanilla with Claude 3 on BookSum. As shown in Figure 2, CoA can outperform the vanilla baseline by a large margin on various source lengths. It is worth noting that, when the length of the sample increases, the performance even increases for CoA, and the improvement over Vanilla (200k) baseline becomes more significant. The improvement of CoA reaches around 100% when the length is larger than 400k. Thus, we can conclude that 1) CoA can still enhance the LLM performance even though the model has a very long context window limit; and 2) CoA delivers more performance gains when the input is longer.

## 5.3 CoA Mitigates "Lost-in-the-Middle" Phenomenon

To assess the "lost-in-the-middle" [42] effect on Vanilla and CoA models, we replicated the original study by randomly selecting 500 samples from their dataset to create a QA dataset. The results are displayed in Figure 4. The Vanilla model exhibits a significant "lost-in-the-middle" issue, with a performance range of 6.13 ($\pm$2.17). In contrast, CoA shows resilience against this issue, with a narrower performance gap of 4.89 ($\pm$1.91), demonstrating that CoA effectively mitigates this problem by providing each agent a shorter context to focus on.

## 5.4 Multi-agent Collaboration in CoA Enables Complex Reasoning over Long Context

Figure 5 displays a sample prediction from HotpotQA. To find the correct answer, RAG retrieves text chunks with high semantic similarity with the *query*. However, conducting multi-hop reasoning is challenging as the critical first-hop *answer* often lacks semantic relevance to the *query*. In contrast, CoA operates differently: the first agent explores related topics without knowing the query's answer, aiding subsequent inference. The second agent, also unaware of the answer, broadens the topic scope by incorporating new information. The third agent finally discovers the answer, synthesizing

Table 7: Ablation on CoA. Manager plays an important role in CoA, and left-to-right yields the best performance among other reading orders including Right-to-Left and Permutation.

| | HotpotQA | MuSiQue | NarrativeQA | Qasper | QuALITY | QMSum | RepoBench-P |
|---|---|---|---|---|---|---|---|
| CoA | 53.62 | **37.09** | **25.26** | **37.17** | **65.42** | **16.77** | 58.25 |
| w/o Manager | 48.58 | 26.79 | 20.80 | 29.66 | 58.80 | 16.50 | 56.16 |
| Right-to-Left | 51.83 | 29.77 | 21.57 | 36.60 | 62.80 | 15.91 | 55.10 |
| Permutation | **56.05** | 34.55 | 23.60 | 37.42 | 64.60 | 16.50 | **58.43** |

Table 8: Comparison of three multi-path augmentation through judge or voting. Multi-path CoA furthers enhance the performance.

| | HotpotQA | MuSiQue | NarrativeQA | Qasper | QuALITY | QMSum | RepoBench-P |
|---|---|---|---|---|---|---|---|
| **Bi-direction: left-to-right and right-to-left paths (2-way)** | | | | | | | |
| w/ judge | 54.11 | 36.97 | 24.47 | 39.23 | 65.00 | 16.41 | 49.69 |
| w/ vote | 51.46 | 34.16 | 21.59 | 37.42 | 64.60 | 9.51 | 38.70 |
| oracle | 62.12 | 48.02 | 32.84 | 46.37 | 71.80 | 18.83 | 59.67 |
| **Self-Consistency: five left-to-right reasoning paths (5-way)** | | | | | | | |
| w/ judge | 57.17 | 38.82 | 21.58 | 36.24 | 62.80 | 17.06 | 46.97 |
| w/ vote | 57.49 | 40.78 | 25.56 | 39.15 | 68.60 | 8.35 | 35.56 |
| oracle | 67.07 | 55.74 | 39.89 | 52.74 | 80.40 | 20.81 | 63.52 |
| **Permutation: five random order reasoning paths (5-way)** | | | | | | | |
| w/ judge | 59.17 | 42.37 | 25.47 | 37.65 | 63.40 | 17.81 | 52.45 |
| w/ vote | 58.29 | 39.17 | 26.58 | 38.09 | 67.60 | 8.31 | 35.44 |
| oracle | 75.73 | 60.16 | 39.58 | 52.22 | 79.80 | 20.88 | 67.80 |

information from earlier agents and new data to complete the reasoning chain. This collaborative approach highlights CoA's ability to facilitate complex reasoning across long context tasks.

## 5.5 Ablation Study: Effectiveness of Manager and Alternative Design Choices

To demonstrate the effect of the manager, we conduct an ablation study that uses the last worker to generate results directly. As shown in Table 7, "w/o Manager" hurts the performance significantly, dropping more than 10% on MuSiQue. This demonstrates the important role of the manager. Next, to empirically verify that left-to-right yields the best performance, we evaluate other orders of reading, including Right-to-Left by reading from the last chunk to the first one and Permutation which reads in random order. As shown in Table 7, on most of the datasets, left-to-right yields the highest score, demonstrating the advantages of natural reading order.

## 5.6 Multi-path Chain-of-Agents Further Enhances Performance

We manually investigated the results over these three orders (left-to-right, right-to-left, permutation), and we found that other orders sometimes can produce better answers than left-to-right. Inspired by this observation, we explore two approaches to select the best result among multiple paths. w/ vote applies majority voting over the final results while w/ judge uses an LLM to judge the most reliable $CU_l$ of diverse paths and generate the final answer. Oracle picks the best path by evaluating score of each path, yielding the upper bound performance. Table 8 compares three multi-path augmentation approaches. Surprisingly, results show that 1) all ensemble approaches (Bi-direction, Self-consistency, and Permutation) can further enhance the performance of CoA and 5-way Permutation yields the best improvement, 2) majority voting (w/ vote) of final answer is better than using an LLM as judge (w/ judge) in Self-consistency, but worse in Bi-direction, 3) using LLM judge (w/ judge) works well on long result generation tasks (QMSum, RepoBench-P), and 4) there is large space to improve because oracle (choose as answer the one with highest performance) is much higher than either w/ judge or w/ vote. We leave the direction of multi-path reasoning to future study.

## 5.7 Practical Time Complexity

We run an analysis to show the practical time consumption of the proposed CoA on HotpotQA dataset. We choose llama3-8b as the backbone model for preventing additional untrackable latency due to the

Table 9: Practical time analysis on HotpotQA dataset. Avg. # of Input/Output shows the total input and output tokens for each model.

| | Running Time (s) | Avg. # of Input | Avg. # of Output | Avg. # Agent Output |
|---|---|---|---|---|
| Vanilla (8k) | 1.33 | 5912.85 | 2.40 | 2.40 |
| RAG (8k) | 2.41 | 16479.91 | 2.75 | 2.75 |
| CoA (8k) | 3.10 | 10840.95 | 38.38 | 11.30 |

Table 10: Information loss of text-bison model on different datasets.

| | HotpotQA | MuSiQue | Qasper | NarrativeQA | QMSum |
|---|---|---|---|---|---|
| CoA performance | 54.87 | 40.38 | 37.03 | 24.04 | 17.01 |
| Information Loss | 1.46 | 1.65 | 3.92 | 3.88 | 0.91 |

network of API queries. As can be seen in the Table 9, Vanilla consumes the least amount of tokens and generates the least of tokens as well. However, it truncates the input thus maintaining a low performance. Although RAG generates fewer tokens than CoA (by adding retrieval and downstream input together), it needs to read the whole input by Retrieval models which also consumes additional time. Overall, RAG is faster than CoA by around 30% in this example. **Parallel decoding.** For decoder-only LLMs, CoA agents can run in parallel. Before CUs are produced, agents can start to read their assigned paragraphs and wait for the CUs to come. We conduct an approximation by asking the model to generate one token for each sample. The output time is shown to be short and negligible, mimicking the encoding time of each sample. We have found that the running time of CoA can be reduced by 57.21% on average, leading to a 1.32-second running time for each sample, closing to the running time of the Vanilla baseline.

## 5.8 Information Loss

To probe the information loss during information propagation in $CU_i$, we further propose a metric to analyze. For each sample, we compute the highest score between the communication unit and the gold answer, and if this score is higher than the score of the final prediction, we compute the difference and refer to this information loss, formulated as $\text{Infomation\_Loss} = \max(0, \max_{i=0}^{l} \text{score}(CU_i, Y) - \text{score}(\hat{Y}, Y))$ where score(*) is the performance metric for each task and dataset, $Y$ is the gold answer and $\hat{Y}$ is the prediction. Table 10 shows the information loss of text-bison. If choosing the communication unit with the highest performance, around 1%-4% performance gain can be obtained, meaning that 1%-4% of the information is lost during the chain communication. While every method might have information loss, CoA design ensures final results yielding negligible information loss.

## 6 Conclusion

In this paper, we propose Chain-of-Agents, a multi-agent LLM collaboration framework for solving long context tasks. It is a training free, task/length agnostic, interpretable, and cost-effective framework. Experiments show that Chain-of-Agents outperforms RAG and Long Context LLMs by a large margin despite of its simple design. Analysis shows that by integrating information aggregation and context reasoning, CoA mitigates lost-in-the-middle and performs better on longer samples.

**Limitations.** While CoA features with a simple and effective design, future directions can address the following limitations to further improve its prowess and efficiency. First, communication effectiveness can be further improved via finetuning or in-context learning because current LLMs are aligned with human norms which is not optimal for communication between LLMs. Second, CoA does not explore other forms of communication approaches, such as debating or complex discussions. Third, the cost and latency of running CoA can be further reduced, such as replacing some LLMs with more effective models via model routing [62].

## Acknowledgement

We thank colleagues in Cloud AI Research team for providing helpful feedback for this paper. We also thank the anonymous reviewers for their helpful comments.

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

# A Proof of Time Complexity

Assuming the source text containing $n$ tokens, window limit of LLM is $k$ tokens, and the responses contain $r$ tokens in average. For decoder-only LLM, we grasp the operations for attention calculation as the time cost unit. Then, for Full-Context LLM, total operation for encoding input source text $T_{\text{Full}}$ is:

$$T_{Enc} = (1 + 2 + \cdots + n) = \frac{(n+1)n}{2} = \mathcal{O}(n^2) \tag{3}$$

Similarly, decoding starts when the model already generate all input. Thus, the first decoded token attends to $n$ positions. Total operation for decoding response is ($r \ll n$):

$$T_{Dec} = (n+1+n+2+\cdots+n+r) = \frac{(n+1+n+r)r}{2} = \mathcal{O}(nr + r^2) = \mathcal{O}(nr) \tag{4}$$

For Chain-of-Agents, we first split the source into $\lceil n/k \rceil$ chunks. Thus, total encoding time for all input is:

$$T_{Enc} = (1 + 2 + \cdots + k) \times \lceil n/k \rceil = \frac{(k+1)k \times \lceil n/k \rceil}{2} = \mathcal{O}(k^2 \times n/k) = \mathcal{O}(nk) \tag{5}$$

Decoding starts when the model already generates $k$ tokens. Thus, the first decoded token attends to $k$ positions. The total operation for decoding response is ($r \ll k$):

$$T_{Dec} = (k+1+k+2+\cdots+k+r) \times \lceil n/k \rceil = \frac{(k+1+k+r)r \times \lceil n/k \rceil}{2} = \mathcal{O}(nr+nr^2/k) = \mathcal{O}(nr) \tag{6}$$

Similarly, for RAG models, encoding time is retrieval time $\mathcal{O}(k'^2 \times n/k') = \mathcal{O}(nk')$ plus the encoding time of downstream LLM $\mathcal{O}(k^2)$. Decoding cost is the time of retrieval embedding generation $\mathcal{O}(n/k')$, plus the generation of the downstream LLM $\mathcal{O}(kr)$.

# B Implementation Details

For all experiments, we use Vertex model garden [4] API to use all six models. Maximum generation token is set to 2048 for gemini-ultra and set to 1024 for the rest of the models. We set temperature to 0 for all experiments except for Self-consistency setting. Table 11 shows the prompt for all models. for task specific requirement of 9 datasets, we follow the original LongBench [6] and SCROLLS [60]. For RAG model, we use the model provided by Huggingface[5] and run on A100 GPUs to rerank the chunks. Algorithm 2 shows the algorithm to chunk the input to cater the window size.

---

**Algorithm 2** Chain of Agents (CoA) Input Chunking Algorithm.

---

**Input:** Source input $x$, query $q$, agent window size $k$, instruction $I_w$.
**Output:** A list of chunks $C = \{c_1, c_2, \cdots, c_l\}$.
   Split the source $x$ into sentences $s_1, s_2, \cdots, s_n$
   Initialize $c \leftarrow$ empty string, $C \leftarrow$ empty ordered set,
   Initialize length budgets $B \leftarrow k - \text{count\_token}(q) - \text{count\_token}(I_w)$.
   **for** $s$ in $s_1, s_2, \cdots, s_n$ **do**
      **if** count_token(c) + count_token(s) > B **then**
         Append $c$ to $C$, initialize $c \leftarrow$ empty string
      **end if**
      $c \leftarrow c \bigoplus s$    // $\bigoplus$ indicates concatenating two strings with a blank.
   **end for**
   **if** count_token(c) $\neq 0$ **then**
      Append $c$ to $C$
   **end if**
   **return** C

---

[4] https://cloud.google.com/model-garden
[5] https://huggingface.co/

Table 11: Prompt of all models for query-based tasks.

| | |
|---|---|
| Vanilla | {Task specific requirement}
{Source Input $x$ with truncation if needed}
Question: {Question $q$}
Answer: |
| RAG | {Task specific requirement}
{Retrieved Chunks of Source Input $x$}
Question: {Question $q$}
Answer: |
| CoA | **Worker $W_i$:**
{Input Chunk $c_i$}
Here is the summary of the previous source text: {Previous Communication Unit ($CU_{i-1}$)}
Question: {Query $q$}
You need to read current source text and summary of previous source text (if any) and generate a summary to include them both. Later, this summary will be used for other agents to answer the Query, if any. So please write the summary that can include the evidence for answering the Query:
**Manager $M$:**
{Task specific requirement}
The following are given passages. However, the source text is too long and has been summarized. You need to answer based on the summary:
{Previous Communication Unit $CU_l$}
Question: {question}
Answer: |

Table 12: Prompt of all models for non-query-based tasks.

| | |
|---|---|
| Vanilla | {Task specific requirement}
{Source Input $x$ with truncation if needed}
Answer: |
| RAG | {Task specific requirement}
{Retrieved Chunks of Source Input $x$}
Answer: |
| CoA | **Worker $W_i$:**
{Input Chunk $c_i$}
Here is the summary of the previous source text: {Previous Communication Unit ($CU_{i-1}$)}
You need to read the current source text and summary of previous source text (if any) and generate a summary to include them both. Later, this summary will be used for other agents to generate a summary for the whole text. Thus, your generated summary should be relatively long.
**Manager $M$:**
{Task specific requirement}
The following are given passages. However, the source text is too long and has been summarized. You need to answer based on the summary:
{Previous Communication Unit $CU_l$}
Answer: |

## C   Example Outputs

We pick one sample from HotpotQA in Table 13, QMSum in Table 14, and RepoBench-P in Table 15 to show the prediction of Chain-of-Agents, in the order of question $q$, Communication Unit $CU$ of agents, and final prediction of manager. We also observe that 1) the first worker usually explores the most number of paths, including 3 or more topics because the agent is not sure about the result and willing to provide more information for the next ones. 2) The final worker usually narrows down to one answer with a much shorter CU, and 3) The speed of narrowing down will be faster for simpler samples, producing a small reasoning graph.

Table 13: Example output on HotpotQA dataset.

| Question | Gary L. Bennett was a part of the space missions that have a primary destination of what celestial body? |
| --- | --- |
| Worker 1 | Gary L. Bennett, a scientist and engineer, has contributed to various space missions, including Voyager, Galileo, and Ulysses. He has worked on advanced space power and propulsion systems and has been involved in planetary protection measures. Bennett's expertise has been instrumental in ensuring the scientific integrity of celestial bodies and preventing harmful contamination. He has received numerous awards and accolades for his contributions to space exploration and is recognized as a leading expert in the field of planetary protection. |
| Worker 2 | Gary L. Bennett, a renowned scientist and engineer, has made significant contributions to space missions, including Voyager, Galileo, and Ulysses. His expertise in advanced space power and propulsion systems, as well as planetary protection measures, has been crucial in ensuring the scientific integrity of celestial bodies and preventing harmful contamination. Bennett has received numerous accolades for his work, including the NASA Exceptional Service Medal and the COSPAR Distinguished Service Award. |
| Worker 3 | Gary L. Bennett, a distinguished scientist and engineer, played a pivotal role in various space missions, particularly Voyager, Galileo, and Ulysses. His expertise in advanced space power and propulsion systems, coupled with his focus on planetary protection measures, has been instrumental in safeguarding the scientific integrity of celestial bodies. Ulysses, launched in 1990, embarked on a unique trajectory to explore both the southern and northern polar regions of the Sun. During its extended mission, Ulysses provided invaluable data on the Sun's magnetic field, solar wind, and the presence of dust in the Solar System. |
| Manager | Sun |
| RAG | Mars |
| Gold | Sun |

Table 14: Example output on QMSum dataset.

| Question | Summarize the discussion about industrial components. |
| --- | --- |
| Worker 1 | The industrial designer proposed that the remote control should be made of titanium or rubber, and the buttons should be simple. The marketing expert said that the remote control should be fancy, innovative, and easy to use. They also discussed the trend of making products look like fruit and vegetables. |
| Worker 2 | The industrial designer proposed that the remote control should be made of titanium or rubber, and the buttons should be simple. The marketing expert said that the remote control should be fancy, innovative, and easy to use. They also discussed the trend of making products look like fruit and vegetables. The industrial designer said that they would explore the two options of titanium and rubber, and see if rubber is expensive. The user interface designer said that they would pretend that the modelling clay is titanium and paint it afterwards. The project manager said that they would stick with two batteries and not reinvent the wheel. |
| Manager | The industrial designer proposed that the remote control should be made of titanium or rubber, and the buttons should be simple. The marketing expert said that the remote control should be fancy, innovative, and easy to use. They also discussed the trend of making products look like fruit and vegetables. The industrial designer said that they would explore the two options of titanium and rubber, and see if rubber is expensive. The user interface designer said that they would pretend that the modelling clay is titanium and paint it afterwards. The project manager said that they would stick with two batteries and not reinvent the wheel. (ROUGE Score: 21.38) |
| RAG | The industrial designer and the marketing manager discussed the industrial components of the remote control. They agreed that the remote control should have a simple design, with a few buttons and a speech recognition technology. They also discussed the possibility of adding a L_C_D_ display, but decided that it would be too expensive. The designer suggested that the remote control could be made of titanium or rubber, and the marketing manager agreed that a spongy design would be more appealing to consumers. (ROUGE Score: 16.54) |
| Gold | The industrial designer provided several options respectively for energy, material and interface. Among these options, the industrial designer preferred traditional battery, titanium and simple push buttons so that they would have enough money for speech recognition. After that, the user interface designer proposed an LCD display only for output, which might be taken into account. The group also talked about how to enable the users to find their controllers. |

Table 15: Example output on RepoBench-P dataset.

| Question | What is the next line of code: … if(type == 5 || type == 0) { |
|---|---|
| Worker 1 | The method SensorDataCollectorManager.flushSensorDataCache(int type, String deviceID) flushes the database cache for the given sensor type. If the type is 0, all sensor types are flushed. The method first checks if the type is valid, and if it is, it calls the flushDBCache method for the appropriate sensor collector. SensorDataCollectorManager.flushSensorDataCache(int type, String deviceID) flushes the database cache for the given sensor type. If the type is 0, all sensor types are flushed. The method first checks if the type is valid, and if it is, it calls the flushDBCache method for the appropriate sensor collector. |
| Worker 2 | The method SensorDataCollectorManager.flushSensorDataCache(int type, String deviceID) flushes the database cache for the given sensor type. If the type is 0, all sensor types are flushed. The method first checks if the type is valid, and if it is, it calls the flushDBCache method for the appropriate sensor collector. |
| Manager | LightSensorCollector.flushDBCache(deviceID); |
| RAG | AccelerometerSensorCollector.flushDBCache(deviceID); |
| Gold | LightSensorCollector.flushDBCache(deviceID); |

Table 16: Compare between CoA and previous state-of-the-art models on nine datasets. * indicates the model needs further training.

|  | HotpotQA | MuSique | Qasper | NarrativeQA | Quality | QMSum | GovReport | BookSum | RepoBench-P |
|---|---|---|---|---|---|---|---|---|---|
| Previous Best | 54.4 [29] | 40.4 [29] | 53.9* [60] | 26 [29] | 89.2 [59] | 22.44* [60] | 26.3 [60] | 18.5* [78] | 56.47 [29] |
| Ours Best | 62.04 | 42.49 | 38.01 | 25.26 | 83.8 | 17.67 | 26.98 | 17.47 | 73.05 |

## D Broader Impacts

Chain-of-Agents is a generic framework for long context tasks. users can apply this to diverse tasks not restricting to the mentioned ones. It will greatly increase the efficiency of individuals or companies to solve complex long context tasks. Besides, the interpretablity of such approach can reduce the misuse of the LLMs because users can check the correctness of results and decrease the possibility of making faults. However, similar to all prompt based approaches, this framework requires careful prompt design for unseen large language models, users may not get optimal solution on certain newly proposed LLMs. Besides, it may increase the number of the calls for API, causing higher network traffic and higher latency for user pools.

## E Comparison with State-of-the-Art Models

Table 16 shows the comparisons of CoA with the previous state-of-the-art performance, including the ones requiring training (indicated with *). As can be seen, CoA achieves better or comparable results on all datasets, improving HotpotQA and RepoBench-p by a large margin. The performance on some datasets is lower than that of state-of-the-art models because the datasets are trained on some domain-specific models, such as Qasper.

## F More Experiments and Analysis

### F.1 Short LLMs on BookSum Dataset

We have run the BookSum dataset with Text-bison and Text-unicorn models. As shown in the Table 17, similar to the results with the LLMs that support longer contexts, CoA outperforms baselines by a significant margin.

### F.2 Robustness against Context Window Size

We set the default context window of CoA to 8k due to the limitation of text-bison and unicorn models. To test the influence of CoA against context window change, we set window sizes to 4k, 8k,

Table 17: Performance of text-bison and text-unicorn on BookSum dataset.

|  | text-bison | text-unicorn |
|---|---|---|
| Vanilla (8k) | 9.13 | 8.15 |
| RAG (8k) | 9.38% | 8.01 |
| CoA (8k) | 14.51 | 14.41 |

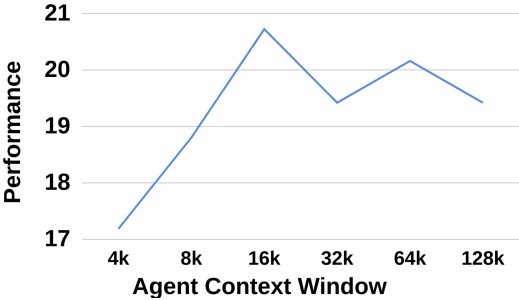

Figure 6: Performance of CoA on Claude 3 Haiku on the NarrativeQA dataset with various context window sizes of an agent. Results show the robustness of CoA towards different choices of context lengths.

16k, 32k, 64k, and 128k of Claude 3 Haiku model and evaluate on NarrativeQA dataset and see the performance change. As shown in Figure 6, the performance of the model increases from 4k to 16k and stabilizes to around 20 with the context window further increases. This result shows that CoA will benefit from increasing length and keep stable when the length touches a bound.

We further analyze different agent lengths and reported the scores of text-bison-32k in Table 18. Our results show that it can improve the baseline with various window sizes. Thus, we choose 8k for short context models, such as text-bison.

### F.3 Evaluation on NIAH Test

We conduct evaluations on the NIAH test. We follow the LongAgent [92] paper to run a NIAH PLUS test for evaluating the long context understanding capability of LLMs. Different from the original NeedleInAHaystack test, NIAH PLUS is more challenging because LLMs need to answer the questions rather than simply retrieve the information (needle). The results are shown in Figure 7. As can be seen, the CoA greatly increases the accuracy of Text-bison from 26.0% to 97.8%, showing that CoA significantly increases the capability of LLMs to understand long contexts.

### F.4 Preventing Collapse of Reasoning Chain

Experiments show that when certain irrelevant content is present in the communication unit $CU_{i-1}$, it often prevents the worker unit $W_i$ from generating a meaningful response. This meaningless response then propagates to the next unit, causing $W_{i+1}$ to produce similarly meaningless content and eventually leading to a collapse of the entire reasoning chain. We call this phenomenon catastrophic collapse. In such cases, the model's overall performance may even fall below that of a single agent $W_1$. A common instance of this phenomenon is a rejection response. If $CU_{i-1}$ includes rejection cues such as "I don't know the answer to the query" or "not mentioned", $W_i$ is likely to repeat

Table 18: Comparison between vanilla and CoA with various context lengths using text-bison-32k.

|  | 4k | 8k | 16k | 32k |
|---|---|---|---|---|
| text-bison-32k | 45.69 | 53.55 | 59.14 | 48.54 |
| Ours (same base) | **54.95** | **60.34** | **63.11** | **50.25** |

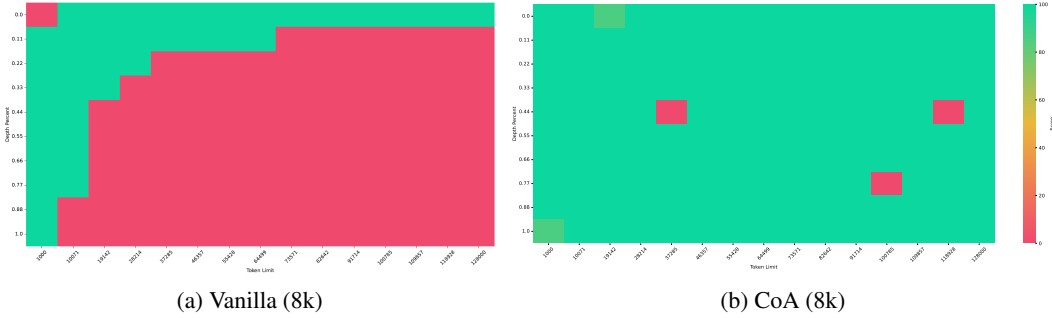

|  (a) Vanilla (8k) | (b) CoA (8k) |
|---|---|

Figure 7: NIAH PLUS test results of Vanilla model and CoA model. Greener/Redder cells indicate higher/lower performance. CoA with 8k improves the performance of the NIAH test greatly compared with the Vanilla baseline.

Table 19: Comparison of related work and their topologies.

| Name or Source | Task or Challenge | Agent Structure | Communication Schema |
|---|---|---|---|
| [14] | Multi-robot planning | Decentralized | Circle |
| RoCo [45] | Multi-robot collaboration | Decentralized | Planning Pipeline |
| CoELA [89] | Multi-Agents cooperation | Decentralized | Memory Module |
| MAD [18] | Improving Factuality | Decentralized | Debate |
| Reconclie [11] | Reasoning | Decentralized | Round Table |
| WalkMaze [9] | Long Input | Centralized | Tree Structure |
| LongAgent [92] | Long Input | Centralized | Tree Structure |
| RecurentGPT [95] | Long Output | Decentralized | Memory Gating |
| CoA (Ours) | Long Input | Decentralized | Chain of Agents |

similar content without adding any meaningful information, often leading to a final output of "not mentioned".

To mitigate this issue, three simple methods can be used: (1) repeatedly running the sample until the final result is not a rejection; (2) postprocessing the communication unit to remove irrelevant words; and (3) using prompt engineering to instruct worker agents to ignore irrelevant information in $CU_i$. We speculate that this issue may be influenced by reinforcement learning from human feedback (RLHF) or chat fine-tuning, causing verbose responses that introduce excessive noise. Further fine-tuning may be needed to address this problem, which we leave to future work.

# G   Detailed Comparison with Related Works

Table 19 illustrates the comparison of related work on multiagent LLM systems. Broadly, prior work on multi-agent LLM systems for long context tasks has a centralized design, where the sibling worker agents do not communicate during inference the communication unit only propagates between manager and worker, and existing decentralized designs do not extend effectively to long context tasks. To the best of our knowledge, CoA is the first to use a decentralized structure on long input tasks and is more effective than baselines such as RAG.

The challenge of long dependency on input remains under-explored (as illustrated in Table 20). Our target is to novelly mitigate challenging context dependency issues in long-context tasks particularly those requiring complex reasoning. To this end, we explored various structures (WalkMaze, Merge, Debate, Group Discussion, etc.) and found that a variant of decentralized well-existing chain communication and this simple approach can work more effectively than others. Our contribution is to mitigate a challenging issue (long input dependency) with a simple intuitive approach (decentralized chain communication). We want to emphasize that centralized communication and our work solve the long context problems in different ways, and they are not in conflict with each other. They can be merged into a much stronger framework (e.g., adding our chain reference stage to mitigate context dependency issues before the stages in LongAgent [92]).

Table 20: Example for comparison of Centralized and Decentralized models.

| Input Source | **Question**: Who is the grandson of A? **Source**: [1],[2],[3],[4] (chunks) **Evidence in each chunk**: [1: A's husband is D], [2: A's son is B], [3: No evidence], [4: B's son is C] |
|---|---|
| Centralized [92] | **Round 1**: Manager: Who is the son of A? Worker with [2]: It is B. Others say unknown **Round 2**: Manager: Who is the son of B? Worker with [4]: It is C. Others say unknown Final answer: It is C. |
| Decentralized (ours) | Manager: Who is the grandson of A? **Worker with [1]**: A's husband is D (topic exploration), **Worker with [2]**: A's son is B (answer first hop), **Worker with [3]**: A's son is B (forward previous evidence), **Worker with [4]**: A's son is B, B's son is C. Thus, A's grandson is C. (complete reasoning) **Final answer**: It is C. |

Table 20 showcases the comparison between CoA and centralized designs in mitigating agent dependency issues for long inputs. Specifically, previous work for long input tasks, despite showing significant improvements, usually uses disentangling algorithms such as question decomposition, to address long-distance dependency in long inputs, and then they use centralized structure to answer the disentangled questions one by one. Different from them, CoA does not disentangle the question but leaves the entire question to the agents. Although chain communication is well-introduced in the literature, to the best of our knowledge, CoA is the first work to apply decentralized communication to long-term dependency challenges. In addition, CoA is not in conflict with existing work as it can be considered as a plugin for other multi-agent LLM systems to improve them.

