# OpenReview forum: "Chain of Agents: Large Language Models Collaborating on Long-Context Tasks"
_NeurIPS.cc/2024/Conference — NeurIPS 2024 poster_

### Official Review · Reviewer_KMmx · 2024-06-19

**Soundness:** 2
**Presentation:** 3
**Contribution:** 2
**Rating:** 5
**Confidence:** 5

**Summary:**

This paper proposes a new method, "Chain-of-Agents" (CoA), to augment the long-context handling capabilities of large language models (LLMs). CoA is a framework designed to enhance the processing of long contexts by sequentially using multiple agents to handle different chunks of input text. In CoA, worker agents manage different segments sequentially and communicate their findings. These findings are then synthesized by a manager agent into a coherent final output, effectively aggregating information and reasoning across the entire context. The author conducted experiments on a wide range of long-context tasks to verify the advantages of CoA.

**Strengths:**

1. The proposed Chain-of-Agents method is an interesting approach, and the authors have experimentally validated its effectiveness.
2. The writing and presentation of the article are great, making it easy to read and follow overall.

**Weaknesses:**

1. My primary concern is the novelty of this submission. Breaking long texts into multiple chunks and processing them sequentially is a well-established practice in both long-content processing and generation. For example, in RecurrentGPT [1], the authors introduced a long-short memory mechanism to store intermediate states during processing, improving the quality of long content generated by LLMs. This is similar to the communication unit (CU) mechanism mentioned in this paper, but the authors do not sufficiently discuss this similarity. Additionally, previous work such as LongAgent [2] proposed segmenting the input long text into several chunks and assigning them to corresponding members. The authors lack sufficient discussion and experimental comparison on utilizing memory mechanisms or agent mechanisms for processing long information.

2. The baselines used in the author's experiments are relatively weak. While the related works section discusses some studies on long text processing, such as references [9] and [13], the implementation process only compares RAG, Vanilla, and COA. The author primarily compares their own designed merge and hierarchical methods in Agent and mechanisms for processing long texts. So, it is challenging to determine the advantages of the proposed methods over existing strong baselines in the field based on the current experiments.

3. Given the existing work on agents collaborating to process multiple chunks, the authors could enhance the novelty of their submission by delving deeper based on the discussion in Section 5.6. Specifically, they could explore the most effective methods of collaboration in multi-chunk processing and agent cooperation. This could add significant technical depth to the paper.


[1] Zhou, Wangchunshu, et al. "Recurrentgpt: Interactive generation of (arbitrarily) long text." arXiv preprint arXiv:2305.13304 (2023).

[2] Zhao, Jun, et al. "LongAgent: Scaling Language Models to 128k Context through Multi-Agent Collaboration." arXiv preprint arXiv:2402.11550 (2024).

**Questions:**

1. How is the chunk split? I did not find a clear description of how the chunk is processed. Could there be a risk of text being abruptly truncated in the middle?

2. As mentioned in the abstract, "input reduction has no guarantee of covering the part with needed information." I am wondering if sequential chunk processing will also encounter these challenges. Could the communication unit potentially drop necessary information if some information is only useful in the context of the following content?

3. In Appendix 1, regarding the calculation of time complexity in lines 592-595, if the model's processing length limit is k and no additional mechanisms are introduced, is the computational complexity of attention calculation O(k^2) or O(n^2)? Please educate me if I miss something important.

4. "a LLM" in line 104 -> "an LLM"

**Limitations:**

The authors adequately addressed the limitations.

---

> ### Author Rebuttal · Authors · 2024-08-07
>
> Thank you for the valuable suggestions. We provide answers to all your weakness points and questions below. We hope these resolve your concerns.
>
> W1: **Novelty and other baselines**:
>
> While chunking the input into multiple segments seems intuitive, the novelty of CoA lies in the chain communication of multiple agents using natural language on inputs, which is not explored by previous studies.
>
> LongAgent [1] does not allow communication between workers, making it hard for each worker to understand what is happening at other agents, hurting the performance of tasks that need long dependency. Besides, the authors of LongAgent state that the decomposition of the question also is a limitation for LongAgents. The failure of decomposition multi-hop or complex questions would directly lead to wrong answers. Additionally, some of the structures of LongAgents are transferable to CoA, such as conflict resolution. In contrast, CoA concentrates on improving the long dependency reasoning (such as multi-hop question answering) and does not depend on the decomposition capability of the manager.
>
> Regarding RecurrentGPT [2], although it uses a chain structure to do long story generation, the task and motivation are different. They focus on memorizing the context to have a better plan to generate a longer story. In contrast, CoA focuses on the long-dependency and complex reasoning of the source text rather than output.
>
> We also compare the performance of CoA with RecurrentGPT and LongAgent, where COA outperforms LongAgent by 16% on NIAH tasks and outperforms RecurrentGPT by 20% on Hotpot QA . More results can be found in GR 2 and Figure 2 in PDF. The results are listed in the Table:
>
> | LongAgent       | Accuracy |
> |-----------------|----------|
> | GPT-4 (128k)    |    62.00 |
> | LongAgent       |    81.53 |
> | Text-bison (8k) |    26.00 |
> | CoA (8k)        |    **97.80** |
>
> | HotpotQA     |       |
> |--------------|-------|
> | RecurrentGPT | 32.54 |
> | CoA (8k)     | **53.62** |
> We will include these discussions and comparisons in our paper.
>
> W2: **Stronger baselines**:
>
>  We believe the baselines used are already strong given the literature in this direction. We are open to benchmarking on any specific suggestions. We have considered other baselines that we modified as they were weaker. For example, WalkingMaze [5] also split the input into chunks; we found it did not work for our task (less than 40% on HotpotQA). Thus, we modify its structure and build a new baseline named Hierarchical which obtains a much stronger performance (50.62% on HotpotQA).
>
> Also, as stated in W1, LongAgents and RecurrentGPT performances are much lower than CoA and thus weaker than the multiagent baseline we have used.
>
> We have further added comparisons of CoA with the previous SOTA below, even including the ones requiring training (indicated with *). As can be seen, CoA achieves better or comparable results on all datasets, improving HotpotQA and RepoBench-p by a large margin. The performance on some datasets is lower than SOTA because the datasets are trained on some domain-specific models, such as Qasper
> |               | HotpotQA | MuSique  | Qasper | NarrativeQA | Quality  | QMsum      | GovReport | BookSum | RepoBench-P |
> |---------------|----------|----------|----------------|-------------|----------|------------|-----------|---------|-------------|
> | Previous Best | 54.4 [1] | 40.4 [1] | 53.9* [2]      | 26 [1]      | 89.2 [3] | 22.44* [2] | 26.3 [3]  | 18.5*[4]   | 56.47 [1]   |
> | Ours Best     |    62.04 |    42.49 |          38.01 |       25.26 |     83.8 |      17.67 |     26.98 |   17.47 |       73.05 |
>
> We will include WalkingMaze, LongAgents,  RecurrentGPT, and other SOTA models in our final paper. Thank you for the great suggestions!
>
> W3: **More agent collaboration**:
>
> We aim to emphasize a simple yet effective framework with strong performance to create a higher impact. It is indeed helpful to include more analysis and another complex framework, we prefer to use this paper to demonstrate the potential of chain communication. As mentioned by Reviewer vGWD and FiRU, being simple and intuitive but effective are the first strengths of the proposed framework.
>
> As described, we have experimented with fairly complex agent collaboration, including bi-directional, permutation, hieratical (multi-layers of information processing), WalkMaze, RecurrentGPT, etc, and the current CoA framework performs the best. In the future, we want to explore more directions, such as high efficiency in communication, complex communication schema, etc.
>
> Q1 **Chunking**: Please refer to the Algorithm in pdf. While there is a small risk of chunking text in the middle, CoA is robust to chunking because the information of previous segments is sent to the next agent for processing, while other baselines such as LongAgents cannot communicate with siblings, losing the context if chunking in the middle.
>
> Q2 **Information loss**: This happens in CoA occasionally and is neglectable.  We have added an analysis to further probe the information loss, as described in W3 of Reviewer FiRU. It shows that COA  ensures final results yielding negligible information loss (only 1%-4%).
>
> Q3 **Time complexity**: It should be O(k^2) because auto-regressive LLMs such as GPT and LLaMA only pay attention to the left side of the input. Thus, each token generated will attend to $k$ tokens [6]. We will clarify this in our paper.
>
>
> [1] Llm maybe longlm: Self-extend llm context window without tuning. ICML  2024.
>
> [2] Scrolls: Standardized comparison over long language sequences. EMNLP 2022.
>
> [3] Zeroscrolls: A zero-shot benchmark for long text understanding. EMNLP Findings 2023
>
> [4] RST-LoRA: A Discourse-Aware Low-Rank Adaptation for Long Document Abstractive Summarization. NAACL 2024.
>
> [5] Walking down the memory maze: Beyond context limit through interactive reading. arXiv, 2023.
>
> [6] Improving language understanding by generative pre-training. arXiv, 2018.

---

> > ### Comment · Reviewer_KMmx · 2024-08-09
> >
> > I appreciate the author's clear and point-by-point rebuttal, as well as the supplementary experiments provided. However, after thoroughly re-examining the paper, the rebuttal, the feedback from the other reviewer, and the related works in this area, I still have **significant concerns** regarding the novelty and the experimental validation of this work.
> >
> > First, as I mentioned in my initial review, the novelty of this submission remains my primary concern. I agree with reviewer *vGWD*'s comment that "*the technical depth of this work is limited.*"
> >
> > Although the authors compare LongAgent and RecurrentGPT in their general response, asserting that "LongAgent **does not allow communication between agents**, making it hard for each agent to understand what is happening with others," various communication mechanisms exist in multi-agent systems (see Section 3.3 in [1] and the vertical and horizontal multi-agent architecture definition in [2]). While CoA uses decentralized communication and LongAgent employs centralized communication, based on my understanding, the central agents in centralized communication essentially function as the communication units described in your paper. Additionally, you mention that LongAgent identifies the decomposition of the input question as a limitation. Could you clarify how CoA addresses this issue? From my understanding, CoA does not involve the decomposition of the input question.
> >
> > Furthermore, as the authors stated, "Regarding RecurrentGPT, although it uses a chain structure to perform long story generation, the task and motivation are different." However, this further diminishes the novelty of the proposed method.
> >
> > In my view, many components of the proposal are common practices in multi-agent system design and long-content generation/processing. The work appears to be a fine-tuned combination of existing methods rather than a deeply innovative approach. In my opinion, the limited novelty does not meet the standards of NeurIPS.
> >
> > Second, regarding the stronger baseline, the authors only tested LongAgent in the NIAH PLUS. Given that LongAgent is closely related to your work and was published three months before your submission, it should be comprehensively tested as a stronger baseline across all benchmarks you tested. Without this, it is difficult to claim that your communication unit or designed agent system is more efficient.
> >
> > Overall, I appreciate the effort the authors have put into the rebuttal. Unfortunately, it does not address my primary concern regarding the work's novelty. I remain open to further discussion with the authors in the coming days.
> >
> > References:
> >
> > [1] Guo, Taicheng, et al. "Large language model based multi-agents: A survey of progress and challenges." arXiv preprint arXiv:2402.01680 (2024).
> >
> > [2] Masterman, Tula, et al. "The landscape of emerging ai agent architectures for reasoning, planning, and tool calling: A survey." arXiv preprint arXiv:2404.11584 (2024).

---

> ### Author Response · Authors · 2024-08-12
> **Reponses from Authors (Part 1)**
>
> Thanks so much for carefully reading our responses, and providing such insightful questions and thoughts! Below, we first re-state our novelty aspects more clearly and then comment on the raised concerns one by one, hoping to make our claims clearer to you and address your concerns!
>
> CoA has a unique positioning in the multi-agent LLM literature. The table below illustrates the comparison of related work and their topologies  (after carefully reading the survey [1], [2], and other literature). Broadly, prior work on multi-agent LLM systems for long input tasks has a centralized design, and existing decentralized designs do not extend effectively to long context tasks. To the best of our knowledge, CoA is the first to use a centralized structure on long input tasks and is more effective than baselines such as WalkMaze and RAG.
> | Work                            | Task                      | Type          | Communication Schema |
> |---------------------------------|---------------------------|---------------|-------------------|
> | [Chen et al., 2023d]           | Multi-robot planning      | Decentralized | Circle            |
> | RoCo [Mandi et al., 2023]       | Multi-robot collaboration | Decentralized | Planning Pipeline |
> | CoELA [Zhang et al., 2023c]     | Multi-Agents cooperation  | Decentralized | Memory Module     |
> | MAD [Du et al., 2023]           | Improving Factuality      | Decentralized | Debate            |
> | Reconclie [Chen et al., 2023]   | Reasoning                 | Decentralized | Round Table       |
> | WalkMaze [Chen et al., 2023]    | Long Input                | Centralized   | Tree Structure    |
> | LongAgent [Zhao et al., 2024]   | Long Input                | Centralized   | Tree Structure    |
> | RecurentGPT [Zhou et al., 2024] | Long Output               | Decentralized |  Memory Gating    |
> | CoA (Ours)                      | Long Input                | Decentralized | Chain of Agents   |
>
> Besides, our contribution is not to claim a new topology for communication. The challenge of long dependency on long input tasks is well-known and remains under-explored (as illustrated with the sample in Part 2). Our target is to *novelly* mitigate **challenging context dependency issues** in long-context tasks particularly those requiring complex reasoning. Thus, it is better if we can find a **simpler yet effective approach**. To this end, we explored various structures (WalkMaze, Merge, Debate, Group Discussion, etc.) and found that a variant of **decentralized well-existing chain communication and this simple approach can work more effectively** than others. Our contribution is not to claim a new topology for communication but to mitigate a challenging issue (long dependency) with a simple intuitive approach (decentralized chain communication). In addition, CoA is not in conflict with existing work as it can be considered as a plugin for other multi-agent LLM systems (e.g. serve as an additional stage in LongAgent) to improve them.
>
> Specifically, previous work for long input tasks, despite showing significant improvements, usually uses disentangling algorithms such as question decomposition, to address long dependency, and then they use centralized structure to answer the disentangled questions one by one. Different from them, CoA does not disentangle the question but leaves the entire question to the agents. Although chain communication is well-introduced in the literature, to the best of our knowledge, CoA is the first work to apply decentralized communication to long-term dependency challenges - please let us know if there is any paper we are missing on this. We show the high effectiveness of CoA in mitigating agent dependency issues for long inputs and we believe it would constitute a significant value add in the multi-agent system literature. We will better clarify these novelty points in the Introduction section of our paper.
>
> Below are the detailed answers to specific points:
>
> > various communication mechanisms exist in multi-agent systems
>
> Indeed, chain communication has been introduced as a very simple topology of communication. Our contribution is not to claim a new topology for communication but to mitigate the challenge of long dependency. We bring a novel perspective to the challenge with the simple chain communication mechanisms. Also, to the best of our knowledge, CoA is the first to use a decentralized structure on long input tasks (see table above). We note that no other multi-agent method in the literature has such results of outperforming RAG for long inputs.

---

> ### Author Response · Authors · 2024-08-12
> **Reponses from Authors (Part 2)**
>
> >  the central agents in centralized communication essentially function as the communication units described in your paper.
>
> Although they all accomplish the task of communication, we think these two types of communication, centralized communication (e.g., LongAgent) and decentralized communication (our CoA), differ in their approach to solving long context dependency issues. When a centralized approach faces a multi-hop question, it leverages question decomposition to disentangle the long dependency issue, while decentralized communication leverages the interleaved reading-processing. Below is an example:
>
> ```
> Question: Who is the grandson of A?
> Source: [1],[2],[3],[4] (chunks)
> Evidence in each chunk: [1: A’s husband is D], [2:  A’s son is B], [3: No evidence], [4: B’s son is C]
> ```
>
> ```
> Centralized communication (e.g., LongAgent):
>
> Round 1:
> Manager: Who is the son of A?
> Worker with [2]: It is B!  Others say unknown
> Round 2:
> Manager: Who is the son of B?
> Worker with [4]: It is C. Others say unknown
> Final answer: It is C.
> ```
>
> In this approach, a worker with [i] and a worker with [i+1] do not communicate. Thus, it is difficult to deal with the dependency when the answer to the question is split into the end of agent i and the start of the agent i+1.
>
> ```
> Decentralized communication (Our CoA):
> Manager: Who is the grandson of A?
> Workers: [1]: A’s husband is D (topic exploration), [2]: A’s son is B (answer first hop), [3]: A’s son is B (forward previous evidence), [4]: A’s son is B, B’s son is C. Thus, A’s grandson is C. (complete reasoning)
> Final answer: It is C
> ```
>
> This exemplifies how adjacent workers communicating in CoA would differ from the centralized one as CoA agents receive previous evidence in addition to the question itself.
>
> We want to emphasize that LongAgent and our work are solving the long context problem in different ways, and they are not in conflict with each other. They can be merged into a much stronger framework (e.g., adding our chain reference stage to mitigate context dependency issues before the stages in LongAgent).
>
> > Could you clarify how CoA addresses this issue? From my understanding, CoA does not involve the decomposition of the input question.
>
> The key aspect of CoA decentralized communication is that workers can communicate so that the question does not need to be decomposed. Our experiments highlight the effectiveness of this design, especially when question decomposition is difficult. For instance, when the passage does not mention “A’s son” directly but mentions “A’s husband is D”, and mentions “D’s son is B”, the first question in the centralized approach should be “Who is A’s husband” rather than “Who is A’s son”, which is difficult to propose.
>
> > However, this further diminishes the novelty of the proposed method.
>
> We acknowledge that chain communication is not first used in our work. But we are the first ones that use chain communication for long dependency in the **input of long context** tasks and we show its effectiveness on generic tasks. This approach is independent of RecurrentGPT, as RecurrentGPT solves **long output**. For the output, the issue they mitigate is memory and planning for story generation tasks. Combining CoA and RecurrentGPT to solve Long-Input-Long-Output generation could be promising as well and we leave that to future work.
>
> > many components of the proposal are common practices in multi-agent system design and long-content generation/processing.
>
> We truly appreciate your thoughtful summary. We understand your concerns and would like to re-emphasize the value of our contributions: our work presents a novel and straightforward solution to the long-standing challenge of long context reasoning, and our results demonstrate significant effectiveness. We are hopeful that our findings will inspire further innovation in the research community, encouraging others to integrate our simple yet powerful components into their future work in multi-agent systems.
>
> > it should be comprehensively tested as a stronger baseline (e.g. LongAgent) across all benchmarks you tested.
>
> Thanks for the suggestion. Indeed, LongAgent is an important work in this direction. We try to compare all datasets in our paper. However, they did not open source their code, making it difficult to reproduce and compare with them fully in this short-time rebuttal window. Since we found that CoA is 15% higher than that of LongAgent (97% vs. 82%) on NIAH PLUS (Figure 2 in pdf), we believe that given the strong improvement, the performance of CoA will be promising. We will include detailed comparisons across all datasets in the final version of our paper.
>
> We hope our response has addressed your concerns and clarified the contributions of our work. We welcome any further questions you may have and would be happy to provide additional clarification if needed. We deeply appreciate the time and effort you've taken to provide feedback on our work.

---

> > ### Comment · Reviewer_KMmx · 2024-08-13
> >
> > Thank you for the clear response, which effectively addresses my concerns regarding how the CoA distinguishes itself from existing multi-agent research. Including these discussions in the paper will enhance its quality and emphasize your contributions. I  would like to raise my score to 5.

---

> > > ### Author Response · Authors · 2024-08-14
> > >
> > > We are really glad that our responses address your concerns! Thanks so much for your thoughtful questions and insightful discussion. We will carefully include our discussion in the final version. We deeply thank the effort you made during the whole process!

---

### Official Review · Reviewer_FfHC · 2024-07-04

**Soundness:** 2
**Presentation:** 2
**Contribution:** 2
**Rating:** 5
**Confidence:** 4

**Summary:**

The paper "Chain of Agents: Large Language Models Collaborating on Long-Context Tasks" introduces a novel framework called Chain-of-Agents (CoA) to address the challenge of effectively processing long contexts in large language models (LLMs). The CoA framework leverages multi-agent collaboration, where multiple worker agents sequentially handle different segments of the text and a manager agent synthesizes these contributions into a coherent final output. This method aims to mitigate the limitations of input reduction and context window extension strategies by enabling effective information aggregation and context reasoning across various LLMs. The framework is evaluated on a range of long-context tasks, including question answering, summarization, and code completion, demonstrating significant improvements over existing methods.

**Strengths:**

- **Task Agnostic and Training Free**: CoA is a versatile framework that does not require task-specific training or fine-tuning, making it applicable to various long-context tasks without additional training overhead.
- **Significant Performance Improvement**: The framework shows significant improvements, up to 10%, over strong baselines like Retrieval-Augmented Generation (RAG) and full-context models in multiple long-context tasks, including question answering, summarization, and code completion.
- **Mitigation of Long-Context Issues**: CoA effectively addresses common issues associated with long contexts, such as the "lost in the middle" phenomenon, by assigning each agent a shorter context, thus maintaining focus on relevant information.

**Weaknesses:**

- **Complexity in Implementation**: Implementing a multi-agent system could be more complex and resource-intensive compared to single-agent systems.
- **Communication Overhead**: The sequential communication between agents might introduce latency and inefficiencies.
- **Evaluation Scope**: While the paper evaluates on multiple datasets, more diverse real-world applications could further validate the robustness of CoA, like NIAH test.

**Questions:**

- Could you test the CoA framework on smaller long-context models, such as qwen2-7b and llama3-8b, which both support an 8k context? I am curious to see how these smart, smaller models perform with CoA.

- Regarding token cost, the input tokens are slightly higher than those in baseline methods. What about the output tokens? How are they related to the number of workers ($l$) and how each worker agent summarizes each chunk?

- In Section 5.2, how do you handle the Full-200k method, which is the Vanilla (200k), when processing contexts over 200k length?

---

> ### Author Rebuttal · Authors · 2024-08-07
>
> Thank you for the valuable feedback. We appreciate your time and efforts spent on this paper. With your insightful suggestions, our paper can improve significantly.
>
> **Complexity in Implementation**
>
> Indeed, one of the design principles behind CoA is to propose a simple yet effective multi-agent system with multiple agents collaborating towards solving a complex task, unlike many other works on multi-agent system design. As can be inferred from the pseudocode in Algorithm 1 in the paper, the proposed approach is very straightforward to implement - with only O(100) lines of code for the end-to-end system. Also, due to its training-free nature, it is not necessary to prepare a training code for it, and the end-to-end system can be effectively operationalized in a highly controllable way. We will also release our code upon acceptance to further help the community implementing the approach. Besides, we compare the time complexity in GR 3, further demonstrating the practicality for deployment, for even scenarios with tight latency budgets.
>
> **Communication Overhead**
>
> We compare the time cost of full-context input and Chain-of Agents theoretically in a decoder-only setting. We assume the response generated by LLMs contains r tokens on average, the input has n tokens, and the context limit of LLM is k. The time complexity is shown in Table 2 (Appendix A).  As can be seen, the encoding time of CoA is less than Full Context because $k <<  n$ in long context tasks, while they have the same decoding time. This demonstrates the efficiency of CoA compared with the Full-Context baseline.
>
>  We have also conducted an experiment of communication cost and latency tests. We run an analysis to show the practical time consumption of the proposed CoA on the HotpotQA dataset. We choose to use LLaMA-3-8b as the backbone model for preventing additional latency due to the network of API queries. As can be seen in the table, Vanilla consumes the least amount of tokens and generates the least of tokens as well. However, it truncates the input thus maintaining a low performance. Although RAG generates fewer tokens than CoA (by adding RAG and downstream input together), it needs to read the whole input by Retrieval models which is also time-consuming. Overall, RAG is faster than CoA by only ~30% in this example.
>
> |              | Running Time (s) | Avg. # of Input  | Avg. # of Output | Avg. # Agent Output |
> |--------------|------------------|--------------|--------------|--------------|
> | Vanilla (8k) |             1.33 |     5,912.85 |         2.40 |         2.40 |
> | RAG (8k)     |             2.41 |    16,479.91 |         2.75 |         2.75 |
> | CoA (8k)     |             3.10 |    10,840.95 |        38.38 |        11.30 |
>
> **Parallel decoding analysis**: For decoder-only LLMs, the agents can run in parallel. Before CUs are produced, agents can start to read their assigned paragraphs and wait for the CUs to come. Unfortunately, current APIs or models do not support such dynamic reading. Thus, we conduct an approximation by asking the model to generate one token for each sample. The output time is shown to be short and negligible, mimicking the encoding time of each sample. We have found that the running time of CoA can be reduced by 57.21% on average, leading to a 1.32-second running time for each sample, closing to the running time of the Vanilla baseline! We will integrate discussions on this speedup approach in the final version of the paper.
>
>
>
>
>
> **Evaluation Scope**:
>
> We have further conducted evaluations on the NIAH test. We follow the LongAgent [1] paper to run a NIAH PLUS test for evaluating the long context understanding capability of LLMs. Different from the original NeedleInAHaystack test, NIAH PLUS is more challenging because LLMs need to answer the questions rather than simply retrieve the information (needle). The results are shown as follows:
>
> | NIAH Test       | Accuracy |
> |-----------------|----------|
> | GPT-4 (128k)    |    62.00 |
> | LongAgent       |    81.53 |
> | text-bison (8k) |    26.00 |
> | CoA (8k)        |    **97.80** |
> As can be seen, the CoA greatly increases the accuracy of Text-bison from 26.0% to 97.8%, showing that CoA significantly increases the capability of LLMs to understand long contexts. We also append the figures of NIAH test results in the attached pdf.
>
> [1] Zhao J, Zu C, Xu H, et al. LongAgent: Scaling Language Models to 128k Context through Multi-Agent Collaboration[J]. arXiv preprint arXiv:2402.11550, 2024.
>
>
> Q1 **Performance on Smaller long context models**: We have added experiments with Llama-3-8b on the NarrativeQA dataset, and the results are listed in the table below. As can be seen, the CoA framework on such LLMs can also boost the Vanilla performance a lot by 10.27% and surpass the RAG score by 5.9%! Noting that there is more space to improve since the prompt of LLama-3 is not even adjusted for this experiment.
>
> | Model        | F1    |
> |--------------|-------|
> | Vanilla (8k) | 8.78 |
> | RAG (8k)     | 13.15 |
> | CoA (8k)     | **19.05** |
>
> Q2 **Output Token Cost**: Regarding output tokens, we compute the average token generated by the model in General Response 3. As shown in the table, CoA outputs more tokens than vanilla and RAG baselines because it needs to produce CUs and final results. We found that the relation between generated tokens and workers holds an almost linear correlation, showing that basically, each worker generates the same amount of tokens. Including one more agent will increase around 11.3 total generated tokens.
>
> Q3 **Full-200k Chunking**: It is the same as Vanilla (8k). Please refer to the Algorithm in pdf. We first split the source into $n$ sentences, then we add $i$-th sentence to the input one by one so that the total length is less than the context window limit while $i+1$ sentences are longer than the total length.

---

> > ### Comment · Reviewer_FfHC · 2024-08-13
> >
> > I would like to thank the authors for the response.
> > My concerns are adequately addressed. Thus I have improved my score to 5.

---

> ### Author Response · Authors · 2024-08-12
>
> Dear Reviewer,
>
> Since we are approaching the end of the discussion period, we are wondering if you have had the chance to review our response to your feedback. We would like to kindly inquire about the extent to which we have successfully addressed the concerns outlined in your review. We greatly value your feedback and would appreciate any further questions or comments you might have.
> Thank you for your time and consideration.
>
> Sincerely,
>
> All Authors

---

> ### Author Response · Authors · 2024-08-14
>
> We are glad to know our responses addressed your concerns! We would also like to thank you for your thoughtful questions and insightful discussion, and we will include all our discussion results in the final version.

---

### Official Review · Reviewer_FiRU · 2024-07-08

**Soundness:** 3
**Presentation:** 3
**Contribution:** 3
**Rating:** 7
**Confidence:** 4

**Summary:**

The paper proposed Chain-of-Agents, a multi-agent LLM collaboration framework for solving long context tasks, where multiple worker agents sequentially comprehend and communicate to handle different segmented portions of the text, and a manager agent, at last, synthesizes these contributions into a coherent final output. The paper conducted experiments on 9 long-context tasks in qa, summarization, and code completion, showing a significant performance gain over vanilla long-context LMs and RAG baselines. The paper also provides abundant analysis.

**Strengths:**

- The proposed method is intuitive.

- The experiments are comprehensive, covering 6 different LLMs as backbones and evaluated across 9 benchmarks.

- The results are good.

- The paper provided useful analysis for more insights.

- The paper is well-written.

**Weaknesses:**

- There is no support in the paper for the claim in L35 "inspired by human-like processing of long-context task", weakening the motivation.

- Results on BookSum are missing in the main table, and no RAG baselines are provided on this benchmark.

- Though the overall results are promising, I'm curious whether there is some information loss during the sequential "read and consume", and how this may affect the performance and the design choice.

- For the case study in Figure 5, for worker 1, there seems to be more than one line of clues to answer the query (a.k.a other space missions such as "Voyager"), which may result in a complicated reasoning graph, how is this phenomenon dealt with the proposed method?

- As also noted in the paper, there is no interactive communication between the agents, but only uni-directional and one-time. Whether the approach could be considered "communication" is debatable.

- Although the time complexity, in theory, is true for Table 2, how's the actual inference speed on GPU considering the overhead of multiple times of extra prompting and decoding?

- It might be helpful to also discuss the related works on communication between language agents, e.g.
  - Camel: Communicative agents for" mind" exploration of large language model society, NeurIPS 23
  - Building Cooperative Embodied Agents Modularly with Large Language Models, ICLR 24

**Questions:**

Please see Weaknesses

**Limitations:**

As the paper also noted, the communication effectiveness and inference efficiency could be further improved.

---

> ### Author Rebuttal · Authors · 2024-08-07
>
> Thank you so much for your detailed feedback and for acknowledging the comprehensiveness of our experimental studies. We address the questions and concerns raised below, one by one.
>
> W1: **Human motivation**: To clarify this point, the motivation is that humans would not try to read the whole textbook and then start to do reasoning. Instead, it is better for them to learn one section and do exercises over it then move to the other one with the memory of the previous section because the human has limited working memory [1], similar to LLMs. This inspires proposing an approach based on interleaved reading, as one of the fundamental principles of CoA, rather than putting all information in one window and processing the information (read-then-process).
>
> W2: **BookSum results**: Thanks for the great suggestion. Following the suggestion, we have run the BookSum dataset with Text-bison and Text-unicorn models. As shown in the table below, similar to the results with the LLMs that support longer contexts, CoA outperforms baselines by a significant margin. We will add these results and discussions to the paper.
>
> |      | text-bison | text-unicorn |
> |------------- |-----------|-------------|
> | Vanilla (8k) |      9.13 |        8.15 |
> | RAG (8k)     |      9.38 |        8.01 |
> | CoA (8k)     |     **14.51** |       **14.41** |
>
>
>  W3: **information loss**: We have added an analysis to further probe the information loss. For each sample, we compute the highest score between the communication unit and the gold answer, and if this score is higher than the score of the final prediction, we compute the difference and refer this information loss, as formulated below:
>
> $$ Loss = max(0,  \max_{i=0}^{l} score(CU_i, Y) - score(\hat{Y}, Y)) $$
>
> The estimated information loss results of various datasets of CoA withext-bison are as follows:
>
> |                  | HotpotQA | MuSiQue | Qasper | NarrativeQA | QMSum  |
> |------------------|----------|---------|--------|-------------|--------|
> | performance      |   54.87 |  40.38 | 37.03 |      24.04 | 17.01 |
> | Information Loss |    1.46 |   1.65 |  3.92 |       3.88 |  0.91 |
>
> This table shows that if choosing the communication unit with the highest performance, around 1%-4% performance gain can be obtained, meaning that 1%-4% of the information is lost during the chain communication. While every method might have information loss, COA design ensures final results yielding negligible information loss.
>
> However, although sometimes the information is lost, "it is still acceptable within the overall framework as we tested using all CUs as the input to the manager and observed a 5% performance drop (Page 4, footnote). We leave mechanisms to further mitigate information loss to future work of multi-agent communication.
>
> W4: **Handling complex reasoning**: We do not give LLMs a predefined plan. When the CoA starts to process, the agents automatically explore the nodes and leave the processed information for the next agent. However, we indeed observed some patterns: 1) the first worker usually explores the most number of paths, including 3 or more topics because the agent is not sure about the result and willing to provide more information for the next ones. 2) The final worker usually narrows down to one answer with a much shorter CU, and 3) The speed of narrowing down will be faster for simpler samples, producing a small reasoning graph.
>
> W5: **Agent communication one directional or more**: Thanks for pointing this out. We agree that we have agent-to-agent communication in one direction and the communication is critical for performance. We will clarify the definition in the next version, such as renaming it as a Unidirectional Communication Unit. Regarding the design choices, we used unidirectional rather than bi-directional (or more complex communication) because experiments show that the bi-directional (or more complex communication) communication of two agents sometimes brings more noise and hallucinations thus hurting the performance.
>
> W6 **Actual inference time**: We further analyze the actual time cost of running samples. Please refer to GR 3 for more details.
>
>
> W7 **Related work discussion**: Thanks. We will add the following papers and compare them with them in the next version.
>
> [1] Cowan N. Working memory underpins cognitive development, learning, and education[J]. Educational psychology review, 2014, 26: 197-223.

---

> ### Author Response · Authors · 2024-08-12
>
> Dear Reviewer,
>
> Since we are approaching the end of the discussion period, we are wondering if you have had the chance to review our response to your feedback. We would like to kindly inquire about the extent to which we have successfully addressed the concerns outlined in your review. We greatly value your feedback and would appreciate any further questions or comments you might have.
> Thank you for your time and consideration.
>
> Sincerely,
>
> All Authors

---

### Official Review · Reviewer_vGWD · 2024-07-13

**Soundness:** 3
**Presentation:** 3
**Contribution:** 3
**Rating:** 5
**Confidence:** 3

**Summary:**

This paper is about addressing the issue of lengthy inputs when using language models. Predominant approach is RAG, but is hampered by retrieval performance. Window extension extends the architecture of the model to handle lengthy inputs, but doesn’t guarantee that the model is able to extract the relevant information from lengthy inputs. This work proposes a simple approach where text is split into multiple chunks and the chunks are sequentially processed, left-to-right, where information from all chunks observed so far is consolidated into a summary. The response to a query (if there is one) is calculated based on the aggregate summary of the text. Despite the simplicity of the approach, the method performs well across various models and tasks.

**Strengths:**

* Idea is quite simple and seems top be effective
* Positive results on many tasks and benchmarks
* It is interesting that even long context models capable of processing lengthy inputs benefit from the proposed method, which reinforces the hypothesis that sifting the relevant information from lengthy inputs directly can be difficult. Consistent improvements are observed on Claude models.
* Appreciate the ablations including effect of direction (left-to-right, right-to-levet) and lost in the middle experiment.
* Seems to work especially well when the inputs are long.

**Weaknesses:**

* Technical depth is limited
* Framing/presentation is somewhat misleading
  - From Figure 1, is summarization all that’s being done? Is CoA is just a fancy term to describe this sequential summarization process?
  - ‘Chain of Agents’ makes it sound like there are different agents doing different tasks, which is misleading
* Fairly local view of performance results. Only comparisons against RAG/Vanilla baselines are presented. How about comparisons to best published numbers on these datasets?
* Clarity issues
  - Figure 1 does not explain the architecture well
  - 127: W1 generates related evidence useful for answering question - Without knowing CU1, the reader cannot verify this. Basically unable to follow the discussion in 126-134 without knowing what CU1,2,3 are.

**Questions:**

* How sensitive is the model to choice of the size of chunks?
* Was the RAG baseline properly optimized?
* Table 2 should also compare against RAG models. For some practical applications, RAG could be more beneficial due to inference speed.

**Limitations:**

Limitations were discussed

---

> ### Author Rebuttal · Authors · 2024-08-07
>
> We thank the reviewer for their insightful feedback that has helped us to improve our submission!
>
> W1: **Technique depth**.
>
> The depth of our work lies in that we propose Chain-of-Agents, a multi-agent LLM collaboration framework for solving long context tasks. It is a training free, task/length agnostic, interpretable, and cost-effective framework. CoA (1) can generalize to many tasks, (2) achieve considerable improvements, (3) while the framework is simple, including diverse communication patterns. Our experiments show that Chain-of-Agents outperforms commonly-used RAG and Long Context LLMs by a large margin despite its simple design.
>
> W2 (1) **Is summarization all that’s being done?**
>
>  We would like to clarify that CoA’s applicability is not restricted to the summarization task. While the communicated results summarize the contents in the source, CoA agents play various roles beyond summarization, because they perform content selection, information aggregation, context utilization, and reasoning abilities to solve complex problems from diverse tasks. such as extracting and refining information through agent communication, and performing analysis and reasoning with them. Tables 11-13 exemplify CoA agents' capabilities to generate evidence for answering the query, a summary of chunks, and code comments and usages.
>
> W2 (2) **Naming as Chain of Agents**:
>
> We name it an \it{agent} for two reasons. First, chain-of-agents contain two different roles, including manager agent and worker agents. Worker agents analyze, aggregate, and digest segment information, whereas manager agents synthesize these contributions into a coherent final output worker. Second, we employ agents to emphasize “communication” between agents. We will clarify this.
>
> W3: **Comparison with SOTA**:
>
> We choose RAG and Full-context as baselines because they represent the commonly-used and solid approaches of input reduction and window extension. Thanks for your great advice. We have further added comparisons of CoA with the previous SOTA below, even including the ones requiring training (indicated with *). As can be seen, CoA achieves better or comparable results on all datasets, improving HotpotQA and RepoBench-p by a large margin. The performance on some datasets is much lower than SOTA because the datasets are trained on some domain-specific models, such as the Qasper dataset
> |               | HotpotQA | MuSique  | Qasper | NarrativeQA | Quality  | QMsum      | GovReport | BookSum | RepoBench-P |
> |---------------|----------|----------|----------------|-------------|----------|------------|-----------|---------|-------------|
> | Previous Best | 54.4 [1] | 40.4 [1] | 53.9* [2]      | 26 [1]      | 89.2 [3] | 22.44* [2] | 26.3 [3]  | 18.5*[4]   | 56.47 [1]   |
> | Ours Best     |    62.04 |    42.49 |          38.01 |       25.26 |     83.8 |      17.67 |     26.98 |   17.47 |       73.05 |
>
> W4 **Clarification of Figure 1**:
>
> We have revised Figure 1 in pdf. The new figure clarifies the working flow and communication units. It is worth noting that the blue boxes on the left are CUs themselves.
>
> Q1 **Context window sizes**:
>
> We show that the CoA becomes stable when the context window is larger than 8k, We also find it can improve the baseline with various window sizes. As shown in Figure 6, the performance becomes stable when it is larger than 8k. We have further analyzed different agent lengths and reported the scores of text-bison on the NarrativeQA dataset in the following table. Our results show that it can improve the baseline with various window sizes. Thus, we choose 8k for short context models, such as text-bison.
>
> | Model\context len | 4k    | 8k    | 16k   | 32k   |
> |-------------------|-------|-------|-------|-------|
> | text-bison-32k    | 45.69 | 53.55 | 59.14 | 48.54 |
> | Ours (same base)  | 54.95 | 60.34 | 63.11 | 50.25 |
>
>
> Q2 **Is RAG well-optimized?**:
>
> The specific RAG [5] implementation we use is indeed well-optimized. It is the SOTA model from the RAG MTEB leaderboard on Hugging Face, and we follow the recent approaches that combine RAG and LLM for the best prompting. Moreover, the RAG model is fine-tuned on the HotpotQA dataset, making it even better adapted for the tasks as the HotpotQA dataset is evaluated in this paper. Overall, CoA outperforming such an optimized RAG implementation highlights the effectiveness.
>
> Q3 **Time complexity analysis**:
>
>  We agree that RAG can be more beneficial due to inference speed. However, it hurts the performance because semantic similarity cannot ensure retrieval of the needed information. It is difficult for RAG models to answer the question that needs multiple reasoning hops (such as multihop-QA) or entire input tokens (such as counting tokens in a passage). We compare the time complexity of RAG with CoA and will add the table in the final version. $k’$ is the chunk size of the RAG model, $k$ is the context window of downstream LLMs, $r$ is the average response length and $n$ is the input source length. We have also compared the time cost and show that RAG is around 30% faster than CoA. Moreover, CoA can be further improved by parallel reading. Please refer to GR 3 for details of inference speed analysis.
>
> |     | Encode          | Decode       |
> |-----|-----------------|--------------|
> | RAG | O(nk') + O(k^2) | O(n) + O(kr) |
> | CoA | O(nk)           | O(nr)        |
>
>
> [1] Jin et al. Llm maybe longlm: Self-extend llm context window without tuning. ICML  2024.
>
> [2] Shaham et al. Scrolls: Standardized comparison over long language sequences. EMNLP 2022.
>
> [3] Shaham et al. Zeroscrolls: A zero-shot benchmark for long text understanding. EMNLP Findings 2023
>
> [4] Pu D, Demberg V. RST-LoRA: A Discourse-Aware Low-Rank Adaptation for Long Document Abstractive Summarization. NAACL 2024.
>
> [5] Xiao S, Liu Z, Zhang P, et al. C-pack: Packaged resources to advance general chinese embedding. SIGIR 2024.

---

> ### Author Response · Authors · 2024-08-12
>
> Dear Reviewer,
>
> Since we are approaching the end of the discussion period, we are wondering if you have had the chance to review our response to your feedback. We would like to kindly inquire about the extent to which we have successfully addressed the concerns outlined in your review. We greatly value your feedback and would appreciate any further questions or comments you might have.
> Thank you for your time and consideration.
>
> Sincerely,
>
> All Authors

---

### Author Rebuttal · Authors · 2024-08-07

We thank all the valuable feedback and comments from reviewers that have helped to improve our paper! We also thank the reviewers for appreciating the intuitive and interesting design of the CoA (Reviewer vGWD, FiRU, FfHC, KMmx), the effectiveness of CoA (Reviewer vGWD, FiRU, FfHC, KMmx), comprehensiveness of the evaluation (Reviewer vGWD, FiRU, FfHC), interesting analysis (Reviewer vGWD, FiRU, FfHC) as well as great presentation and writing (Reviewer FiRU, KMmx).

We address several key questions in the following paragraphs:

GR 1 **Novelty and Technical Depth**

While chunking the input into multiple segments seems intuitive, the novelty of CoA lies in the chain communication of multiple agents using natural language on inputs, which is not explored by previous studies, demonstrating that such a simple yet effective framework with strong performance can create a higher impact.

LongAgent does not allow communication between agents, making it hard for each agent to understand what is happening at others, hurting the performance of tasks that need long dependency. Besides, the authors of LongAgent state that the decomposition of the input question also is a limitation for LongAgents. The failure of decomposition multi-hop or complex questions would directly lead to wrong answers. Additionally, some of the structures of LongAgents are transferable to CoA, such as conflict resolution. In contrast, CoA concentrates on improving the long dependency reasoning (such as multi-hop question answering) and does not depend on the decomposition capability of the manager.

Regarding RecurrentGPT, although it uses a chain structure to do long story generation, the task and motivation are different. They focus on memorizing the context to have a better plan to generate a longer story.  In contrast, CoA focuses on the long-dependency and complex reasoning of the source text rather than generation.

The depth of our work lies in that we propose Chain-of-Agents, a multi-agent LLM collaboration framework for solving long context tasks. It is a training-free, task/length agnostic, interpretable, and cost-effective framework. While the framework is conceptually straightforward, CoA can (1) generalize to many tasks, (2) achieve considerable improvements, and (3) demonstrate diverse communication patterns. Our experiments show that CoA outperforms commonly-used RAG and Long Context LLMs by a large margin despite its simple design. Analysis shows that by integrating information aggregation and context reasoning, CoA mitigates lost-in-the-middle phenomenon effectively, and performs better on longer samples.


GR 2 **Evaluation Scope**

We used nine representative, diverse datasets for three broad categories of long context tasks, and we believe the baselines used are already strong given the literature in this direction. As described in individual responses, we will include comparisons with more baselines including WalkingMaze, LongAgents, RecurrentGPT, and other SOTA models in our final paper. These additional results further corroborate consistent and significant performance improvements of CoA.

We follow the LongAgent paper to run a NIAH PLUS test for evaluating the long context understanding capability of LLMs. Different from the original NeedleInAHaystack test, NIAH PLUS is more challenging because LLMs need to answer the questions rather than simply retrieve the information (needle). The results are shown as follows:

| NIAH PLUS Test  | Accuracy |
|-----------------|----------|
| GPT-4 (128k)    |    62.00 |
| LongAgent       |    81.53 |
| text-bison (8k) |    26.00 |
| CoA (8k)        |    **97.80** |
As can be seen, CoA greatly increases the accuracy of Text-bison from 26.0% to 97.8%, showing that CoA significantly increases the capability of LLMs to understand long contexts.  We also append the figures of NIAH test results in the attached pdf.



GR 3 **Practical Time Complexity**

We run an analysis to show the practical time consumption of the proposed CoA on HotpotQA dataset. We choose to use LLaMA-3-8b as the backbone model for preventing additional latency due to the network of API queries. As can be seen in the table, Vanilla consumes the least amount of tokens and generates the least of tokens as well. However, it truncates the input thus maintaining a low performance. Although RAG generates fewer tokens than CoA (by adding RAG and downstream input together), it needs to read the whole input by Retrieval models which is also time-consuming. Overall, RAG is faster than CoA by only ~30% in this example.

|              | Running Time (s) | Avg. # of Input  | Avg. # of Output | Avg. # Agent Output |
|--------------|------------------|--------------|--------------|--------------|
| Vanilla (8k) |             1.33 |     5,912.85 |         2.40 |         2.40 |
| RAG (8k)     |             2.41 |    16,479.91 |         2.75 |         2.75 |
| CoA (8k)     |             3.10 |    10,840.95 |        38.38 |        11.30 |

**Parallel decoding analysis**: For decoder-only LLMs, CoA agents can run in parallel. Before CUs are produced, agents can start to read their assigned paragraphs and wait for the CUs to come. Unfortunately, current APIs or models do not support such dynamic reading. Thus, we conduct an approximation by asking the model to generate one token for each sample. The output time is shown to be short and negligible, mimicking the encoding time of each sample. We have found that the running time of CoA can be reduced by **57.21%** on average, leading to a 1.32-second running time for each sample, closing to the running time of the Vanilla baseline! We will integrate discussions on this speedup approach in the final version of the paper.

---

### Decision · Program_Chairs · 2024-09-25

**Decision:**

Accept (poster)

**Comment:**

In this work, the authors propose Chain-of-Agents (CoA), a training-free framework that leverages multi-agent collaboration through natural language to solve long context tasks. Specifically, CoA first split long text into multiple chunks, then each text chunk is processed in a sequential manner (from left to right). At any step, with the awareness of the query, information from all chunks observed so far is consolidated into a summary, and being passed along to the agent processing the next chunk. Finally, a manager agent performs a final information integration and subsequently generates the response.

This work has many merits. Reviewers agree that the idea is intuitive, simple yet effective; the method is task agnostic and training free, makes it accessible; the authors conducted a comprehensive set of experiments, on 9 benchmarks with 6 LLM backbones, CoA show strong experimental results on all tasks, this further highlighted the accessibility of the work; reviewer find the authors' analysis useful and insightful; they find the paper well-written, with clear presentation; they believe the work provides interesting insights, the method can be empirically impactful. The all-positive scores suggest the reviewers' overall sentiment to this work.

Before the author-reviewer discussion period, reviewers' concerns were mainly about:
- It was unclear how this work connects/differs from prior works such as LongAgent and RecurrentGPT where they also have similar system design that breaks long text into multiple chunks.
- Reviewers felt unsatisfied about the baselines because the work only compared with RAG and Vanilla LLMs, stronger baselines were missing.
- The details of the agent communication was unclear.
- The inference speed/latency/efficiency was unclear.
- There lacks of evidence how CoA could generalize to smaller/weaker LLMs (with long context support).

The author did a great job addressing the concerns and answering the questions during the discussion period, they successfully convinced reveiwer FfHC to raise their score from 4 to 5; and reviewer KMmx to raise their score from 3 to 5. They both acknowledge that the authors had addressed their concerns effectively/adequately.

During the discussion period, the authors made their work stronger, thanks to the reveiwers' suggestions:
- It is much clearer now why CoA is unique among prior/concurrent work that has similar design. The table provided in their response to Reviewer KMmx is very helpful, and should be included in their cameraready.
- The additional experiments using LLaMA-3-8b as backbone made it more convincing that CoA can be applied to open-sourced models.
- The authors additionally compared their work on all the 9 datasets with previous best systems, they show that CoA could achieve better or comparable results on all datasets (comparable even with prior systems that require additional in-domain training).
- The authors additonally tested CoA against a set of baselines on an extreme test, the Needle in a Haystack (NIAH) test set, as suggested by Reviewer FfHC. CoA achieves near perfect performance and outperforms other systems by a large margin.

Overall, I believe if the authors could well integrate the additional clarification and experiments/analyses into their cameraready, this work could be well received by the NeurIPS community. Although this work presents a training-free method (which I personally less in favor of), but I like the fact the method is straightforward and effective, plus the authors did a good job justifying how this simple method could be applied to a potentially wide scenario, I do see good empirical impact this work could offer.